# ALLEVIATING PRIVACY ATTACKS VIA CAUSAL LEARNING

## ABSTRACT

Machine learning models, especially deep neural networks have been shown to reveal membership information of inputs in the training data. Such *membership inference* attacks are a serious privacy concern, for example, patients providing medical records to build a model that detects HIV would not want their identity to be leaked. Further, we show that the attack accuracy amplifies when the model is used to predict samples that come from a different distribution than the training set, which is often the case in real world applications. Therefore, we propose the use of *causal* learning approaches where a model learns the causal relationship between the input features and the outcome. An ideal causal model is known to be invariant to the training distribution and hence generalizes well to shifts between samples from the same distribution and across different distributions. First, we prove that models learned using causal structure provide stronger differential privacy guarantees than associational models under reasonable assumptions. Next, we show that causal models trained on sufficiently large samples are robust to membership inference attacks across different distributions of datasets and those trained on smaller sample sizes always have lower attack accuracy than corresponding associational models. Finally, we confirm our theoretical claims with experimental evaluation on 4 datasets with moderately complex Bayesian networks. We observe that neural network-based associational models exhibit upto 80% attack accuracy under different test distributions and sample sizes whereas causal models exhibit attack accuracy close to a random guess. Our results confirm the value of the generalizability of causal models in reducing susceptibility to privacy attacks.

## 1 INTRODUCTION

Machine learning algorithms, especially deep neural networks (DNNs) have found diverse applications in various fields such as healthcare (Esteva et al., 2019), gaming (Mnih et al., 2013), and finance (Tsantekidis et al., 2017; Fischer & Krauss, 2018). However, a line of recent research has shown that deep learning algorithms are susceptible to privacy attacks that leak information about the training dataset (Fredrikson et al., 2015; Rahman et al., 2018; Song & Shmatikov, 2018; Hayes et al., 2017). Particularly, one such attack called *membership inference* reveals whether a particular data sample was present in the training dataset (Shokri et al., 2017). The privacy risks due to membership inference elevate when the DNNs are trained on sensitive data such as in healthcare applications. For example, patients providing medical records to build a model that detects HIV would not want to reveal their participation in the training dataset.

Membership inference attacks are shown to exploit overfitting of the model on the training dataset (Yeom et al., 2018). Existing defenses propose the use of generalization techniques such as adding learning rate decay, dropout or using adversarial regularization techniques (Nasr et al., 2018b; Salem et al., 2018). All these approaches assume that the test data is from the same distribution as the training dataset. In practice, a model trained using data from one distribution is often used on a (slightly) different distribution. For example, hospitals in one region may train a model to detect HIV and share it with hospitals in different regions. However, generalizing to a new context is a challenge for any machine learning model. We extend the scope of membership privacy to different distributions and show that the risk from membership attack increases further on DNNs as the test distribution is changed. That is, the abiltity of an adversary to distinguish a member from a non-member improves with change in test distributions.

To alleviate privacy attacks, we propose using models that depend on the causal relationship between input features and the output. Causal learning has been extensively used to guarantee fairness and

explainability properties of the predicted output (Kusner et al., 2017; Nabi & Shpitser, 2018; Datta et al., 2016). However, the connection of causal learning to privacy is yet unexplored. To the best of our knowledge, we provide the first analysis of privacy benefits of causal models. By definition, causal relationships are invariant across input distributions (Peters et al., 2016), and hence make the predictions of *causal models* independent of the observed data distribution, let alone the observed dataset. Hence, causal models generalize better even with change in the distributions.

In this paper, we show that the generalizability property of causal models directly ensures better privacy guarantees for the input data. Concretely, we prove that with reasonable assumptions, a causal model always provides stronger (i.e., smaller $\epsilon$ value) differential privacy guarantees than a corresponding associational model trained on the same features and the same amount of added noise to the training dataset. Consequently, we show that membership inference attacks are ineffective (equivalent to a random guess) on causal models trained on infinite samples. Empirical attack accuracies on four different datasets confirm our theoretical claims. We find that 60K training samples are sufficient to reduce the attack accuracy of a causal model to a random guess. In contrast, membership attack accuracy for neural network-based associational models increase as test distributions are changed. The attack accuracy reaches nearly 80% when the target associational model is trained on 60K training samples and used to predict test data that belong to a different distribution than the training data. Our results show that causal learning approaches are a promising direction for training models on sensitive data. Section 2 describes the properties of causal models. Section 3 proves the connections of causality to differential privacy and robustness to membership attacks. Section 4 provides empirical results. To summarize, we make the following contributions:

- For the same amount of added noise, models learned using causal structure provide stronger $\epsilon$-differential privacy guarantees than corresponding associational models.
- Causal models are *provably* more robust to membership inference attacks than typical associational models such as neural networks.
- We simulate practical settings where the test distribution may not be the same as the training distribution and find that the membership inference attack accuracy of causal models is close to a "random guess" (i.e., 50%) while associational models exhibit upto $80\%$ attack accuracy.

## 2 Properties of Causal Models

Causal models are shown to generalize well since the output of these models depend only on the causal relationship between the input features and the outcomes instead of the associations between them. From prior work, we know that the causal relationship between the features is invariant to the their distribution (Peters et al., 2016). Using this property, we study its effects on the privacy of data.

### 2.1 Background: Causal Model

Intuitively, a causal model identifies a subset of features that have a causal relationship with the outcome and learns a function from the subset to the outcome. To construct a causal model, one may use a structural causal graph based on domain knowledge that defines causal features as parents of the outcome under the graph. Alternatively, one may exploit the strong relevance property from Pellet & Elisseeff (2008), use score-based learning algorithms (Scutari, 2009) or recent methods for learning invariant relationships from training datasets from different distributions (Peters et al., 2016; Bengio et al., 2019), or learn based on a combination of randomized experiments and observed data. Note that this is different from training probabilistic graphical models, wherein an edge conveys an associational relationship. Further details on causal models are in Pearl (2009); Peters et al. (2017).

For ease of exposition, we assume the structural causal graph framework throughout. Consider data from a distribution $(X, Y) \sim P$ where $X$ is a $k$-dimensional vector and $Y \in \{0, 1\}$. Our goal is to learn a function $h(X)$ that predicts $Y$. Figure 1 shows causal graphs that denote the different relationships between $X$ and $Y$. Nodes of the graph represent variables and a directed edge represents a direct causal relationship from a source to target node. Denote $X_{pa} \subseteq X$, the parents of $Y$ in the causal graph. Figure 1a shows the scenario where $X$ contains variables $X_{S0}$ that are correlated to $X_{pa}$ in $P$, but not necessarily connected to either $X_{pa}$ or $Y$. These correlations may change in the future, thus a generalizable model should not include these features. Similarly, Figure 1b shows parents and children of $X_{pa}$. The d-separation principle states that a node is independent of its ancestors conditioned on all its parents (Pearl, 2009). Thus, $Y$ is independent of $X_{S1}$ and $X_{S2}$ conditional on $X_{pa}$. Therefore, including them in a model does not add predictive value (and further, avoids problems when the relationships between $X_{S1}$ and $X_{S2}$ may also change). Finally, for

completeness, the exhaustive set of variables to include is known as the *causal Markov Blanket*[1], $X_C$ which includes $Y$'s parents, ($X_{pa}$), children ($Y_{ch}$) and parents of children. Conditioned on its Markov blanket (Figure 1c), $Y$ is independent of all other variables in the causal graph. When $Y$ has no descendants in the graph, then the effective Markov blanket includes only its parents, $X_{pa}$.

The key insight is that building a model for predicting $Y$ using the Markov Blanket $X_C$ ensures that the model generalizes to other distributions of $X$, and also to changes in other causal relationships between $X$, as long as the causal relationship of $X_C$ to $Y$ is stable. We call such a model as a *causal* model, the features in ($X_C$) as the *causal features*, and assume that all the causal features for $Y$ are observed. In contrast, we call a model that uses all available features as an *associational* model.

Figure 1: A causal predictive model includes only the parents of $Y$ (a) and (b). Panel (c) shows the generalization to a Markov Blanket.

## 2.2 GENERALIZATION TO NEW DISTRIBUTIONS

We state the generalization property of causal models and show how it results in a stronger differential privacy guarantee. We first define `In-distribution` and `Out-of-distribution` generalization error. Throughout, $L(.,.)$ refers to the loss on a single input and $\mathcal{L}_P(.,.) = \mathbb{E}_P L(.,.)$ refers to the expected value of the loss over a distribution $P(X, Y)$. We refer $f : X \to Y$ as the ground-truth labeling function and $h : X \to Y$ as the hypothesis function or simply the model. Then, $L(h, h')$ is any loss function quantifying the difference between two models $h$ and $h'$.

**Definition 1. In-Distribution Generalization Error** (`IDE`). *Consider a dataset $S \sim P(X, Y)$. Then for a model $h : X \to Y$ trained on $S$, the in-distribution generalization error is given by:*

$$\text{IDE}_P(h, f) = \mathcal{L}_P(h, f) - \mathcal{L}_{S \sim P}(h, f) \tag{1}$$

**Definition 2. Out-of-Distribution Generalization Error** (`ODE`). *Consider a dataset $S$ sampled from a distribution $P(X, Y)$. Then for a model $h : X \to Y$ trained on $S$, the out-of-distribution generalization error with respect to another distribution $P^*(X, Y)$ is given by:*

$$\text{ODE}_{P,P^*}(h, f) = \mathcal{L}_{P^*}(h, f) - \mathcal{L}_{S \sim P}(h, f) \tag{2}$$

**Definition 3. Discrepancy Distance** (`disc`$_L$) (**Def. 4 in Mansour et al. (2009)**). *Let $\mathcal{H}$ be a set of hypotheses, $h : X \to Y$. Let $L : Y \times Y \to \mathbb{R}_+$ define a loss function over $Y$ for any such hypothesis $h$. Then the discrepancy distance $\text{disc}_L$ over any two distributions $P(X, Y)$ and $P^*(X, Y)$ is given by:*

$$\text{disc}_L(P, P^*) = \max_{h, h' \in \mathcal{H}} |\mathcal{L}_P(h, h') - \mathcal{L}_{P^*}(h, h')| \tag{3}$$

Intuitively, the term $\text{disc}_L(P, P^*)$ denotes the distance between the two distributions. Higher the distance, higher is the chance of an error when transferring $h$ from one distribution to another. Now, we will state the theorem on the generalization property of causal models.

**Theorem 1.** *Consider a structural causal graph $G$ that connects $X$ to $Y$, and causal features $X_C$ where $X_C$ is a Markov Blanket of $Y$ under $G$. Let $P(X, Y)$ and $P^*(X, Y)$ be two distributions with arbitrary $P(X)$ and $P^*(X)$ such that the causal relationship between $X_C$ and $Y$ is preserved, which implies that $P(Y|X_C) = P^*(Y|X_C)$. Let $f : X_C \to Y$ be the resultant invariant labelling function such that $y = f(X_C)$. Further, assume that $\mathcal{H}_C$ represents the set of causal models $h_c : X_C \to Y$ that use all causal features and $\mathcal{H}_A$ represent the set of associational models $h_a : X \to Y$ that may use all available features, such that $\mathcal{H}_C \subseteq \mathcal{H}_A$ and $f \in \mathcal{H}_C$.*

*Then, for any symmetric loss function $L$ that obeys the triangle inequality, the upper bound of `ODE` from a dataset $S \sim P(X, Y)$ to $P^*$(called `ODE-Bound`) for a causal model $h_c \in \mathcal{H}_C$ is less than or equal to the upper bound `ODE-Bound` of an associational model $h_a \in \mathcal{H}_A$, with probability at least $(1 - \delta)^2$.*

$$\text{ODE-Bound}_{P,P^*}(h_c, f; \delta) \leq \text{ODE-Bound}_{P,P^*}(h_a, f; \delta) \tag{4}$$

---

[1]We call it the *causal* Markov Blanket since it is based on the structural causal graph, to distinguish it from the associational Markov Blanket that is based on conditional probability distribution from a Bayesian network.

*Proof.* As an example, consider a colored MNIST data distribution $P$ where the classification task is to detect whether a digit is greater than $5$, and where all digits above $5$ are colored with the same color. Then, under a suitably expressive class of models, the loss-minimizing associational model may use only the color feature to obtain zero error, while the loss-minimizing causal model will still use the shape (causal) features to obtain zero error. On any new $P^*$ that does not follow the same correlation of digits with color, we expect that the loss-minimizing associational model will have higher error than the loss-minimizing causal model.

Formally, since $P(Y|X_c) = P^*(Y|X_c)$, the optimal causal model that minimizes loss over $P$ is the same as the loss-minimizing model over $P^*$. That is, $h_{c,P}^{OPT} = h_{c,P^*}^{OPT}$. However, for some associational models, $h_{a,P}^{OPT} \neq h_{a,P^*}^{OPT}$ and thus there is an additional loss term when generalizing to data from $P^*$. The rest of the proof follows from triangle inequality of the loss function and the standard bounds for IDE from past work. Detailed proof is in Appendix Section A.1. □

**Corollary 1.** *Consider a causal model $h_c : X_C \to Y$ and an associational model $h_a : X \to Y$ trained on a dataset $S \sim P$. Let $(x, y) \in S$ and $(x', y') \notin S$ be two input instances such that they share the same true labelling function $y = f(x_c)$ and $y' = f(x'_c)$. Then, the worst-case generalization error for a causal model on any such $x'$ is less than or equal to that for an associational model.* [Proof in Appendix Section A.2]

$$\max_{x \in S, x':y'=f(x'_c)} L_{x'}(h_c, f) - L_x(h_c, f) \leq \max_{x \in S, x':y'=f(x'_c)} L_{x'}(h_a, f) - L_x(h_a, f) \tag{5}$$

## 3 MAIN RESULT: PRIVACY GUARANTEES WITH CAUSALITY

We now present our main result on the privacy guarantees and attack robustness of causal models.

### 3.1 DIFFERENTIAL PRIVACY GUARANTEES

Differential privacy (Dwork et al., 2014) provides one of the strongest notion of privacy guarantees to hide the participation of an individual sample in the dataset. To state informally, it ensures that the presence or absence of a single data point in the input dataset does not change the output by much.

**Definition 4** (Differential Privacy). *A mechanism M with domain $\mathcal{I}$ and range $\mathcal{O}$ satisfies $\epsilon$-differential privacy if for any two datasets $d, d' \in \mathcal{I}$ that differ only in one input and for a set $\mathcal{S} \subseteq \mathcal{O}$, the following holds:* $\Pr(\mathcal{M}(d) \in \mathcal{S}) \leq e^{\epsilon} \Pr(\mathcal{M}(d') \in \mathcal{S})$

Based on the generalization property, we show that causal models provide stronger differential privacy guarantees than corresponding associational models. The standard approach to designing a differentially private algorithm is by calculating the sensitivity of that algorithm and adding noise proportional to the sensitivity. Sensitivity captures the change in the output of an algorithm due to the change in a single data point in the input. Higher the sensitivity, larger is the amount of noise required to make an algorithm differentially private with reasonable $\epsilon$ guarantees. We first provide the formal definition of sensitivity and then show that the sensitivity of causal models is lower than or equal to associational models.

**Definition 5** (Sensitivity (From Def. 3.1 in Dwork et al. (2014))). *Let $\mathcal{F}$ be a function that maps a dataset to a vector in $\mathbb{R}^d$. Let $S, S'$ be two datasets such that $S'$ differs from $S$ in one data point. Then the $l_1$-sensitivity of a function $\mathcal{F}$ is defined as:* $\Delta\mathcal{F} = \max_{S,S'} ||\mathcal{F}(S) - \mathcal{F}(S')||_1$

**Lemma 1.** *Let $S$ be a dataset over $(X, Y)$ values, such that $y^{(i)} = f(x_c^{(i)}) \, \forall (x^{(i)}, y^{(i)}) \in S$, where $f$ is the true labelling function over the causal features $X_C$. Consider a neighboring dataset $S'$ such that $S' = S \backslash (x, y) + (x', y')$ where $(x, y) \in S$, $(x', y') \notin S$, and $(x', y')$ shares the same causal labelling function $y' = f(x'_c)$. Let a model $h$ be specified by a set of parameters $\theta \in \Omega \subseteq \mathbb{R}^n$. Let $h_S^{min}(x; \theta_S)$ be a model learnt using $S$ as training data and $h_{S'}^{min}(x; \theta_{S'})$ be the model learnt using $S'$ as training data. Then, the sensitivity of a causal learning function $\mathcal{F}_c$ that outputs learnt empirical hypothesis $h_{c,S}^{min} \leftarrow \mathcal{F}_c(S)$ and $h_{c,S'}^{min} \leftarrow \mathcal{F}_c(S')$ is lower than or equal to the sensitivity of an associational learning function $\mathcal{F}_a$ that outputs $h_{a,S}^{min} \leftarrow \mathcal{F}_a(S)$ and $h_{a,S'}^{min} \leftarrow \mathcal{F}_a(S')$ using a loss function $L$ that is strongly convex over $\Omega$, symmetric and obeys the triangle inequality,*

$$\Delta\mathcal{F}_c = \max_{S,S'} ||h_{c,S}^{min} - h_{c,S'}^{min}||_1 \leq \max_{S,S'} ||h_{a,S}^{min} - h_{a,S'}^{min}||_1 = \Delta\mathcal{F}_a \tag{6}$$

*where the maximum is over all such datasets $S$ and $S'$.*

*Proof.* We can write the empirical loss minimizers for the datasets $\mathtt{S}$ and $\mathtt{S}'$ as:

$$\mathtt{h_S^{min}} = \arg\min_{\mathtt{h}} \mathcal{L}_{\mathtt{S}}(\mathtt{h}, \mathtt{f}) = \arg\min_{\mathtt{h}} \frac{1}{\mathtt{N}} \sum_{\mathtt{i}=1}^{\mathtt{N}} \mathtt{L_{x_i}}(\mathtt{h}, \mathtt{f})$$

$$\mathtt{h_{S'}^{min}} = \arg\min_{\mathtt{h}} \mathcal{L}_{\mathtt{S}'}(\mathtt{h}, \mathtt{f}) = \arg\min_{\mathtt{h}} \frac{1}{\mathtt{N}} \sum_{\mathtt{i}=1}^{\mathtt{N}-1} \mathtt{L_{x_i}}(\mathtt{h}, \mathtt{f}) + [\mathtt{L_{x'}}(\mathtt{h}, \mathtt{f}) - \mathtt{L_x}(\mathtt{h}, \mathtt{f})]$$

(7)

From Corollary 1, for $\mathtt{x}' \notin \mathtt{S}$ and $\mathtt{x} \in \mathtt{S}$, we have:

$$\max_{\mathtt{x},\mathtt{x}'} \mathtt{L_{x'}}(\mathtt{h_c}, \mathtt{f}) - \mathtt{L_x}(\mathtt{h_c}, \mathtt{f}) \leq \max_{\mathtt{x},\mathtt{x}'} \mathtt{L_{x'}}(\mathtt{h_a}, \mathtt{f}) - \mathtt{L_x}(\mathtt{h_a}, \mathtt{f})$$

(8)

Since $\mathtt{x} \in \mathtt{S}$ and $\mathtt{x}' \in \mathtt{S}'$ and $|\mathtt{S} - \mathtt{S}'| = 1$ the above is true for any $\mathtt{S}$ and $\mathtt{S}'$,

$$\max_{\mathtt{S},\mathtt{S}'} \mathcal{L}_{\mathtt{S}'}(\mathtt{h_c}, \mathtt{f}) - \mathcal{L}_{\mathtt{S}}(\mathtt{h_c}, \mathtt{f}) \leq \max_{\mathtt{S},\mathtt{S}'} \mathcal{L}_{\mathtt{S}'}(\mathtt{h_a}, \mathtt{f}) - \mathcal{L}_{\mathtt{S}}(\mathtt{h_a}, \mathtt{f})$$

(9)

Since $\mathtt{L}$ is a strongly convex function over $\Omega$, and since $\mathcal{H}_C \subseteq \mathcal{H}_A \Rightarrow \Omega_C \subseteq \Omega_A$, $\mathtt{h_{c,S}^{min}}$ and $\mathtt{h_{c,S'}^{min}}$ that minimize $\mathcal{L}_{\mathtt{S}}(\mathtt{h}, \mathtt{f})$ and $\mathcal{L}_{\mathtt{S}'}(\mathtt{h}, \mathtt{f})$ respectively should also be closer to each other than $\mathtt{h_{a,S}^{min}}$ and $\mathtt{h_{a,S'}^{min}}$ (Boyd & Vandenberghe, 2004) (using Eqn. 9).

$$\max_{\mathtt{S},\mathtt{S}'} || \theta_{\mathtt{c,S}}^{\mathtt{min}} - \theta_{\mathtt{c,S'}}^{\mathtt{min}} ||_1 \leq \max_{\mathtt{S},\mathtt{S}'} || \theta_{\mathtt{a,S}}^{\mathtt{min}} - \theta_{\mathtt{a,S'}}^{\mathtt{min}} ||_1 \Rightarrow \max_{\mathtt{S},\mathtt{S}'} || \mathtt{h_{c,S}^{min}} - \mathtt{h_{c,S'}^{min}} ||_1 \leq \max_{\mathtt{S},\mathtt{S}'} || \mathtt{h_{a,S}^{min}} - \mathtt{h_{a,S'}^{min}} ||_1 \quad (10)$$

Hence, sensitivity of a causal model is lower than an associational model i.e., $\Delta\mathcal{F}_c \leq \Delta\mathcal{F}_a$. $\qquad\square$

**Theorem 2.** *Let $\hat{\mathcal{F}}_c$ and $\hat{\mathcal{F}}_a$ be the differentially private algorithms corresponding to causal learning and associational learning algorithms $\mathcal{F}_c$ and $\mathcal{F}_a$ respectively. Let $\hat{\mathcal{F}}_c$ and $\hat{\mathcal{F}}_a$ provide $\epsilon_c$-DP and $\epsilon_a$-DP guarantees respectively. Then, for noise sampled from the same distribution, $\mathtt{Lap(Z)}$ for both algorithms, we have $\epsilon_c \leq \epsilon_a$.*

*Proof.* According to the Def. 3.3 of Laplace mechanism from Dwork et al. (2014), we have,

$$\hat{\mathcal{F}}_c = \mathcal{F}_c + \mathcal{K} \sim \mathtt{Lap}(\frac{\Delta\mathcal{F}_c}{\epsilon_c}) \qquad \hat{\mathcal{F}}_a = \mathcal{F}_a + \mathcal{K} \sim \mathtt{Lap}(\frac{\Delta\mathcal{F}_a}{\epsilon_a})$$

(11)

The noise is added to the output of the learning algorithm i.e., the model parameters. Since $\mathcal{K}$ is sampled from the same noise distribution,

$$\mathtt{Lap}(\frac{\Delta\mathcal{F}_c}{\epsilon_c}) = \mathtt{Lap}(\frac{\Delta\mathcal{F}_a}{\epsilon_a}) \qquad \therefore \frac{\Delta\mathcal{F}_c}{\epsilon_c} = \frac{\Delta\mathcal{F}_a}{\epsilon_a}$$

(12)

From Lemma 1, $\Delta\mathcal{F}_c \leq \Delta\mathcal{F}_a$ and hence $\epsilon_c \leq \epsilon_a$. $\qquad\square$

While we prove the general result above, our central claim comparing differential privacy for causal and associational models also holds true for models developed using recent work (Papernot et al., 2017) that provides a tighter data-dependent differential privacy guarantee. The key idea is to produce an output label based on voting from $M$ teacher models, each trained on a disjoint subset of the training data. We state the theorem below and provide the proof in Appendix B. Given datasets from different domains, the below theorem provides a constructive proof to generate a differentially private causal algorithm, following the method from Papernot et al. (2017).

**Theorem 3.** *Let $\mathtt{D}$ be a dataset generated from possibly a mixture of different distributions $\mathtt{P(X, Y)}$ such that $\mathtt{P(Y|X_C)}$ remains the same. Let $n_j$ be the votes for the jth class from M teacher models. Let $\mathcal{M}$ be the mechanism that produces a noisy max, $\arg\max_j\{n_j + Lap(2/\gamma)\}$. Then the privacy budget $\epsilon$ for a causal model is lower than that for the associational model with the same accuracy.*

### 3.2 Robustness to Membership Attacks

Deep learning models have shown to memorize or overfit on the training data during the learning process (Carlini et al., 2018). Such overfitted models are susceptible to *membership inference attacks* that can accurately predict whether a target input belongs to the training dataset or not (Shokri et al., 2017). There are multiple variants of the attack depending on the information accessible to the

adversary. An adversary with access to a black-box model only sees the confidence scores for the predicted output whereas one with the white-box has access to the model parameters and observe the output at each layer in the model (Nasr et al., 2018a). In the black-box setting, a membership attack is possible whenever the distribution of output scores for training data is different from the test data, and has been connected to model overfitting (Yeom et al., 2018). For the white-box setting, if an adversary knows the true label for the target input, then they may guess the input to be a member of the training set whenever the loss is lower, and vice-versa. Alternatively, if the adversary knows the distribution of the training inputs, they may learn a "shadow" model based on synthetic inputs and use the shadow model's output to build a membership classifier for any new input (Salem et al., 2018).

Most of the existing membership inference attacks have been demonstrated for test inputs from the same data distribution as the training set. When test inputs are expected from the same distribution, methods to reduce overfitting (such as adversarial regularization) can help reduce privacy risks (Nasr et al., 2018b). However, in practice, this is seldom the case. For instance, in our example of a model trained to detect HIV, the test inputs may come from different hospitals. Models trained to reduce the generalization error for a specific test distribution are still susceptible to membership inference when the distribution of features is changed. This is due to the problem of *covariate shift* that introduces a domain adaptation error term (Mansour et al., 2009). That is, the loss-minimizing model that predicts $Y$ changes with a different distribution, and thus allows the adversary to detect differences in losses for the test versus training datasets. As we show below, causal models alleviate the risk of membership inference attacks. From Yeom et al. (2018), we first define a membership attack as:

**Definition 6.** *Let model $h$ be trained on a dataset $S(X, Y)$ of size $N$. Let $\mathcal{A}$ be an adversary with access to $h$ and a input $X$. The advantage of an adversary in membership inference is the difference between true and false positive rate in guessing whether the the input belongs to the training set.*

$$\text{Adv}(\mathcal{A}, h) = \Pr[\mathcal{A} = 1 | b = 1] - \Pr[\mathcal{A} = 1 | b = 0] \tag{13}$$

*where $b = 1$ if the input is in the training set and $0$ otherwise.*

**Lemma 2.** *[From Yeom et al. (2018)] Let $\mathcal{M}$ be a $\epsilon$-differentially private mechanism based on a model $h$. The membership advantage for an adversary is bounded by $\exp(\epsilon) - 1$.*

**Theorem 4.** *Given a structural causal network that connects $X$ to $Y$, let $S \sim P(X, Y)$ be a dataset sampled from $P$, and let $P^*$ be any distribution such that $P(Y|X_C) = P^*(Y|X_C)$. Then, a causal model $h_c$ trained on $S$ yields lower membership advantage than an associational model $h_a$ even when the test dataset is from a different distribution $P^*$.*

*Proof.* From Theorem 2 above, we can construct an $\epsilon_c$-DP mechanism based on a causal model, and a $\epsilon_a$-DP mechanism based on an associational model, where $\epsilon_c \leq \epsilon_a$. Further, this construction works for different input distributions. From Lemma 2, the membership advantage of an adversary $\mathcal{A}$ is,

$$\text{Adv}(\mathcal{A}, h_c) \leq \exp(\epsilon_c) - 1 \qquad\qquad \text{Adv}(\mathcal{A}, h_a) \leq \exp(\epsilon_a) - 1 \tag{14}$$

Thus, worst case advantage for a causal model is always lower than that of an associational model. $\square$

**Corollary 2.** *Let $h_{c,S}^{\min}$ be a causal model trained using empirical risk minimization on a dataset $S \sim P(X, Y)$ with sample size $N$. As $N \to \infty$, membership advantage $\text{Adv}(\mathcal{A}, h_{c,S}^{\min}) \to 0$.*

The proof is the based on the result from Theorem 1 that $h_{c,P}^{\text{OPT}} = h_{c,P^*}^{\text{OPT}}$ for a causal model. Crucially, membership advantage does not go to zero as $N \to \infty$ for associational models, since $h_{a,P}^{\text{OPT}} \neq h_{a,P^*}^{\text{OPT}}$ in general. Detailed proof is in Appendix Section C.

**Attribute Inference attacks.** We prove similar results on the benefits of causal models for attribute inference attacks in Appendix Section D.

## 4 IMPLEMENTATION AND EVALUATION

**Benchmark Datasets.** To avoid errors in learning causal structure from data, we perform evaluation on datasets for which the causal structure and the true conditional probabilities of the variables are known

Table 1: Details of the benchmark datasets

| Dataset | Child | Alarm | (Sachs) | Water |
|---|---|---|---|---|
| Output | XrayReport | BP | Akt | CKNI_12_45 |
| No. of classes | 5 | 3 | 3 | 3 |
| Nodes | 20 | 37 | 11 | 32 |
| Arcs | 25 | 46 | 17 | 66 |
| Parameters | 230 | 509 | 178 | 10083 |

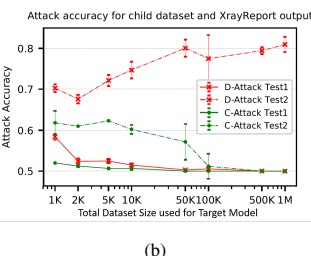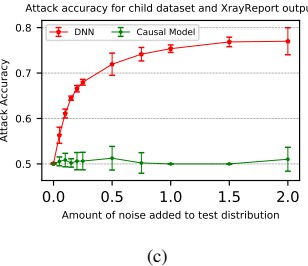

Figure 2: Results for Child dataset with XrayReport as the output. ( a) is the target model accuracy. ( b) is the attack accuracy for different dataset sizes on which the target model is trained and ( c) is the attack accuracy for test distribution with varying amount of noise for total dataset size of 100K samples.

from prior research. We select 4 Bayesian
network datasets— Child, Sachs, Alarm and Water that range from 230-10k parameters (Table 1) (bnl). Nodes represent the number of input features and arcs denote the causal connections between these features in the network. Each causal connection is specified using a conditional probability table $P(X_i|\texttt{Parents}(X_i))$; we consider these probability values as the parameters in our models. To create a prediction task, we select a variable in each of these networks as the output $Y$. The number of classes in Table 1 denote all the possible values for an output variable. For example, the variable BP (blood pressure) in the alarm dataset takes 3 values i.e, `LOW, NORMAL, HIGH`. The causal model uses only parents of $Y$ whereas the associational model (DNN) uses all nodes except $Y$ as features.

**Implementation.** We sample data using the causal structure and probabilities from the Bayesian network, and use a 60:40% split for train-test datasets. We learn a causal model and a deep neural network (DNN) on each training dataset. We implement the attacker model to perform membership inference attack using the output confidences of both these models, based on past work (Salem et al., 2018). The input features for the attacker model comprises of the output confidences from the target model, and the output is membership prediction (member / non-member) in the training dataset of the target model. In both the train and the test data for the attacker model, the number of members and non-members are equal. The creation of the attacker dataset is described in Figure 4 in Appendix. Note that the attack accuracies reported are an upper bound since we assume that the adversary has white-box access to the ML model.

To train the causal model, we use the bnlearn library in R language that supports maximum likelihood estimation of the parameters in $Y$'s conditional probability table. For prediction, we use the `parents` method to predict the class of any specific variable. To train the DNN model and the attacker model, we build custom estimators in Python using Tensorflow v1.2 ten. The DNN model is a multilayer perceptron (MLP) with 3 hidden layers of 128, 512 and 128 nodes respectively. The learning rate is set to 0.0001 and the model is trained for 10000 steps. The attacker model has 2 hidden layers with 5 nodes each, a learning rate of 0.001, and is trained for 5000 steps. Both models use Adam optimizer, ReLU for the activation function, and cross entropy as the loss function. We chose these parameters to ensure model convergence.

### 4.1 EXPERIMENTAL SETUP

We evaluate the DNN and the causal model sample sizes ranging from 1K to 1M dataset sizes. We refer Test 1 as the test dataset which is drawn from the same distribution as the training data and Test 2 is generated from a completely different distribution except for the relationship of the output class to its parents. To generate Test 2, we alter the true probabilities $\Pr(X)$ uniformly at random (later, we consider adding noise to the original value). Our goal with generating Test 2 is to capture the realistic behaviour of shift in distribution for input features. We refer the causal and DNN model as the *target* on which the attack is perpetrated.

### 4.2 RESULTS

We present results on the accuracy of target models (causal and DNN models) and the membership attack accuracy for different dataset sizes and test distributions.
**Accuracy comparison of DNN and Causal models.** Figure 2a shows the target model accuracy comparison for the DNN and the causal model trained on the Child dataset with XrayReport as the output variable. We report the accuracy of the target models only for a single run since in practice the attacker would have access to the outputs of only a single model. We observe that the DNN model

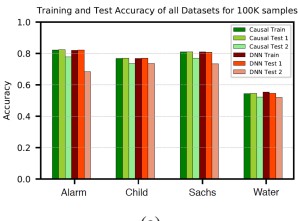 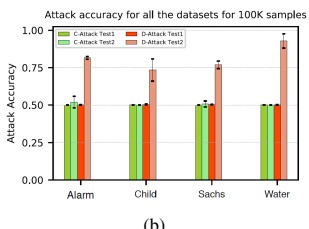 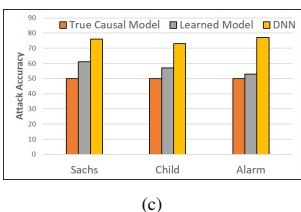

Figure 3: (a)Target accuracy, (b) Attack accuracy, (c) Attack accuracy for true, learned causal model and DNN.

has a large difference between the train and the test accuracy (both Test 1 and Test 2) for smaller dataset sizes (1K and 2K). This indicates that the model overfits on the training data for these dataset sizes. However, after 10K samples, the model converges such that the train and Test 1 dataset have the same accuracy. The accuracy for the Test 2 distribution stabilizes for a total dataset size of 10K samples. In contrast, for the causal model, the train and Test 1 accuracy are similar for the causal model even on smaller dataset sizes. However, after convergence at around 10K samples, the gap between the accuracy of train and Test 2 dataset is the same for both the DNN and the causal model. Figure 3a shows the accuracy comparison for all four datasets with similar results.

**Attack Accuracy comparison of DNN and Causal models.** A naive attacker classifier would predict all the samples to be members and therefore achieve 0.5 prediction accuracy. Thus, we consider 0.5 as the baseline attack accuracy which is equal to a random guess. Figure 2b shows the attack accuracy comparison for Test 1 (same distribution) and Test 2 (different distribution) datasets. Attack accuracy of the Test 1 dataset for the causal model is slightly above a random guess for smaller dataset sizes, and then converges to 0.5. In comparison, attack accuracy for the DNN on Test 1 dataset is over 0.6 for smaller samples sizes and reaches 0.5 after 10K datapoints. This confirms past work that an overfitted DNN is susceptible to membership inference attacks even for test data generated from the same distribution as the training data (Yeom et al., 2018). On Test 2, the attack accuracy is always higher for the DNN than the causal model, indicating our main result that associational models "overfit" to the training distribution, in addition to the training dataset. Membership inference accuracy for DNNs is as high as 0.8 for total dataset size of 50K while that of causal models is below 0.6. Further, attack accuracy for DNN increases with sample size whereas attack accuracy for the causal model reduces to 0.5 for total dataset size over 100k even when the gap between the train and test accuracies is the same as DNNs ( as shown in Figure 2a). These results show that causal models generalize better than DNNs across input distributions. Figure 3b shows a similar result for all four datasets. The attack accuracy for DNNs and the causal model is close to 0.5 for the Test 1 dataset while for the Test 2 dataset the attack accuracy is significantly higher for DNNs than causal model. This empirically confirms our claim that in general, causal models are robust to membership inference attacks across test distributions as compared to associational models.

**Attack Accuracy for Different Test Distributions.** To understand the change in attack accuracy as $\Pr(X)$ changes, we also generate test data from different distributions by adding varying amount of noise to the true probabilities. We range the noise value between 0 to 2 and add it to the individual probabilities which are then normalized to sum up to 1. Figure 2c shows the comparison of attack accuracy for the causal model and the DNN on the child dataset for a total sample size of 100K samples. We observe that the attack accuracy increases with increase in the noise values for the DNN. Even for a small amount of noise, attack accuracies increase sharply. In contrast, attack accuracies stay close to 0.5 for the causal model, demonstrating the robustness to membership attacks.

**Results with learnt causal model.** Finally, we perform experiments to understand the effect of privacy guarantees on causal structures learned from data that might be different from the true causal structure. We evaluate the attack accuracy for learned causal models on the `Sachs, Child` and `Alarm` dataset[2]. For these datasets, a simple hill-climbing algorithm returned the true causal parents. Hence, we evaluated attack accuracy for models with hand-crafted errors in learning the structure i.e., misestimation of causal parents, see Figure 3c. Specifically, we include two non-causal features as parents of the output variable along with the true causal features. The attack risk increases as a learnt model deviates from the true causal structure, however it still exhibits lower attack accuracy than the corresponding associational model. Table 2 in Appendix gives a fine-grained analysis.

---

[2]We exclude the `Water` dataset as `bn.fit` in bnlearn library gives error due to the extreme probabilities.

## 5    RELATED WORK

**Privacy attacks and defenses on ML models.**  Shokri et al. (2017) demonstrate the first membership inference attacks on black box neural network models with access only to the confidence values. Similar attacks have been shown on several other models such as GANs (Hayes et al., 2017), text prediction generative models (Carlini et al., 2018; Song & Shmatikov, 2018) and federated learning models (Nasr et al., 2018b). However, prior research does not focus on the severity of these attacks with change in the distribution of the test dataset. We discussed in Section 2 that existing defenses based on regularization (Nasr et al., 2018b) are not practical when models are evaluated on test inputs from different distributions. Another line of defense is to add differentially private noise while training the model. However, the $\epsilon$ values necessary to mitigate membership inference attacks in deep neural networks require addition of large amount of noise that degrades the accuracy of the output model (Rahman et al., 2018). Thus, there is a trade-off between privacy and utility when using differential privacy for neural networks. In contrast, we show that causal models require lower amount of noise to achieve the same $\epsilon$ differential privacy guarantees and hence retain accuracy closer to the original model. Further, as training sample sizes become sufficiently large, as shown in Section 4, causal models are, by definition, robust to membership inference attacks across distributions.

**Causal learning and privacy.** There is a substantial literature on learning causal models from data; for a review see (Peters et al., 2017; Pearl, 2009). Kusner et al. (2015) proposed a method to privately reveal parameters from a causal learning algorithm, using the framework of differential privacy. Instead of a specific causal algorithm, our focus is on the privacy benefits of causal models for general predictive tasks. While recent work applies causal models to study properties of machine learning models such as providing explanations (Datta et al., 2016) or fairness (Kusner et al., 2017), the relation of causality to privacy is yet unexplored. With this paper, we present the first result which shows the privacy benefits of causal models.

## 6    CONCLUSION AND FUTURE WORK

We conclude that causal learning is a promising approach to train models which are robust to privacy attacks such as membership inference and model inversion. As our future work, we want to relax our assumption of a known causal structure and investigate the privacy guarantees of causal models where the causal features and the relationship between them is not known apriori (Peters et al., 2017).

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

# A GENERALIZATION PROPERTIES OF CAUSAL MODEL

## A.1 GENERALIZATION OVER DIFFERENT DISTRIBUTIONS

**Theorem 1.** *Consider a structural causal graph $G$ that connects $X$ to $Y$, and causal features $X_C$ where $X_C$ is a Markov Blanket of $Y$ under $G$. Let $P(X, Y)$ and $P^*(X, Y)$ be two distributions with arbitrary $P(X)$ and $P^*(X)$ such that the causal relationship between $X_C$ and $Y$ is preserved, which implies that $P(Y|X_C) = P^*(Y|X_C)$. Let $f : X_C \to Y$ be the resultant invariant labelling function such that $y = f(X_C)$. Further, assume that $\mathcal{H}_C$ represents the set of causal models $h_c : X_C \to Y$ that use all causal features and $\mathcal{H}_A$ represent the set of associational models $h_a : X \to Y$ that may use all available features, such that $\mathcal{H}_C \subseteq \mathcal{H}_A$ and $f \in \mathcal{H}_C$.*

*Then, for any symmetric loss function $L$ that obeys the triangle inequality, the upper bound of $ODE$ from a dataset $S \sim P(X, Y)$ to $P^*$(called $ODE{-}Bound$) for a causal model $h_c \in \mathcal{H}_C$ is less than or equal to the upper bound $ODE{-}Bound$ of an associational model $h_a \in \mathcal{H}_A$, with probability at least $(1 - \delta)^2$.*

$$ODE{-}Bound_{P,P^*}(h_c, f; \delta) \leq ODE{-}Bound_{P,P^*}(h_a, f; \delta) \tag{4}$$

*Proof.* Consider a model $h : X \to Y$ belonging to a set of models $\mathcal{H}$, that was trained on $S \sim P(X, Y)$. From Def. 2 we write,

$$\begin{aligned} ODE_{P,P^*}(h, f) &= \mathcal{L}_{P^*}(h, f) - \mathcal{L}_{S\sim P}(h, f) \\ &= \mathcal{L}_{P^*}(h, f) - \mathcal{L}_P(h, f) + \mathcal{L}_P(h, f) - \mathcal{L}_{S\sim P}(h, f) \\ &= \mathcal{L}_{P^*}(h, f) - \mathcal{L}_P(h, f) + IDE_P(h, f) \end{aligned} \tag{15}$$

where the last equation is to due to Def.1 of the in-distribution generalization error.

Let us denote the optimal loss-minimizing hypotheses over $\mathcal{H}$ for $P$ and $P^*$ as $h_P^{OPT}$ and $h_{P*}^{OPT}$.

$$h_P^{OPT} = \arg\min_{h \in \mathcal{H}} \mathcal{L}_P(h, f) \qquad h_{P*}^{OPT} = \arg\min_{h \in \mathcal{H}} \mathcal{L}_{P*}(h, f) \tag{16}$$

Using the triangle inequality of the loss function, we can write:

$$\mathcal{L}_{P*}(h, f) \leq \mathcal{L}_{P*}(h, h_P^{OPT}) + \mathcal{L}_{P*}(h_P^{OPT}, h_{P*}^{OPT}) + \mathcal{L}_{P*}(h_{P*}^{OPT}, f) \tag{17}$$

And,

$$\begin{aligned} \mathcal{L}_P(h, f) &\geq \mathcal{L}_P(h, h_P^{OPT}) - \mathcal{L}_P(h_P^{OPT}, f) \\ \Rightarrow -\mathcal{L}_P(h, f) &\leq -\mathcal{L}_P(h, h_P^{OPT}) + \mathcal{L}_P(h_P^{OPT}, f) \end{aligned} \tag{18}$$

Thus, combining Eq. 15, 17 and 18, we obtain,

$$\begin{aligned} ODE_{P,P^*}(h, f) &\leq IDE_P(h, f) + \mathcal{L}_{P*}(h, h_P^{OPT}) + \mathcal{L}_{P*}(h_P^{OPT}, h_{P*}^{OPT}) + \mathcal{L}_{P*}(h_{P*}^{OPT}, f) - \mathcal{L}_P(h, h_P^{OPT}) + \mathcal{L}_P(h_P^{OPT}, f) \\ &= IDE_P(h, f) + (\mathcal{L}_{P*}(h, h_P^{OPT}) - \mathcal{L}_P(h, h_P^{OPT})) + \mathcal{L}_{P*}(h_{P*}^{OPT}, f) + \mathcal{L}_P(h_P^{OPT}, f) + \mathcal{L}_{P*}(h_P^{OPT}, h_{P*}^{OPT}) \\ &\leq IDE_P(h, f) + disc_{L,\mathcal{H}}(P, P^*) + \mathcal{L}_{P*}(h_{P*}^{OPT}, f) + \mathcal{L}_P(h_P^{OPT}, f) + \mathcal{L}_{P*}(h_P^{OPT}, h_{P*}^{OPT}) \end{aligned} \tag{19}$$

where the last inequality is due to the definition of discrepancy distance (Definition 3). Equation 19 divides the out-of-distribution generalization error of a hypothesis $h$ in four parts:

1. $\text{IDE}_\text{P}(\text{h}, \text{f})$ denotes the in-distribution error of $\text{h}$. This can be bounded by typical generalization bounds, such as the uniform error bound that depends only on the VC dimension and sample size of $\mathcal{S}$ (Shalev-Shwartz & Ben-David, 2014). Using a uniform error bound based on the VC dimension, we obtain, with probability at least $1 - \delta$,

$$\text{IDE} \leq \sqrt{8 \frac{\text{VCdim}(\mathcal{H})(\ln(2|\mathcal{S}|) + 1) + \ln(4/\delta)}{|\mathcal{S}|}} = \text{IDE-Bound}(\mathcal{H}, \mathcal{S}) \qquad (20)$$

Since $H_C \subseteq H_A$, VC-dimension of causal models is not greater than that of associational models. Thus,

$$\text{VCDim}(\mathcal{H_C}) \leq \text{VCDim}(\mathcal{H_A}) \Rightarrow \text{IDE-Bound}(\mathcal{H_C}, \mathcal{S}) \leq \text{IDE-Bound}(\mathcal{H_A}, \mathcal{S}) \qquad (21)$$

2. $\text{disc}_{\text{L},\mathcal{H}}(\text{P}, \text{P}^*)$ denotes the distance between the two distributions. Given two distributions, the discrepancy distance does not depend on $\text{h}$, but only on the hypothesis class $\mathcal{H}$.
3. $\mathcal{L}_{\text{P}^*}(\text{h}_{\text{P}^*}^{\text{OPT}}, \text{f})$ and $\mathcal{L}_\text{P}(\text{h}_\text{P}^{\text{OPT}}, \text{f})$ measure the error due to the true labeling function $\text{f}$ being outside the hypothesis class $\mathcal{H}$.
4. $\mathcal{L}_{\text{P}^*}(\text{h}_\text{P}^{\text{OPT}}, \text{h}_{\text{P}^*}^{\text{OPT}})$ denotes the loss (or difference) between the loss-minimizing function over $\text{P}$ and the loss-minimizing hypothesis over $\text{P}^*$.

We next consider two classes of models, $\mathcal{H_C}$ that contains models that uses all causal features ($\text{X}_\text{C}$) from a Markov Blanket over the structural causal graph, and $\mathcal{H_A}$ that contains associational models that may use all or a subset of all available features. Below we show that for a given distribution $\text{P}$ and another distribution $\text{P}^*$ such that $\text{P}(\text{Y}|\text{X}_\text{C}) = \text{P}^*(\text{Y}|\text{X}_\text{C})$, the last term $\mathcal{L}_\text{P}(\text{h}_\text{P}^{\text{OPT}}, \text{h}_{\text{P}^*}^{\text{OPT}})$ of Equation 19 vanishes for a causal model but may remain non-zero for an associational model.

**Causal Model.** Given a structural causal network, let us construct a model using all variables $\text{X}_\text{C}$ in $\text{Y}$'s Markov Blanket. By property of the structural causal network, $\text{X}_\text{C}$ includes all parents of $\text{Y}$ and thus there are no backdoor paths, using Rule 2 of do-calculus from Pearl (2009):

$$\Pr(\text{Y}|\text{do}(\text{X}_\text{c} = \text{x}_\text{c})) = \text{P}(\text{Y}|\text{X}_\text{C} = \text{x}_\text{c}) = \text{P}^*(\text{Y}|\text{X}_\text{C} = \text{x}_\text{c}) \qquad (22)$$

where the last equality is since data from $\text{P}^*$ also shares the same causal graph, ($\text{Y}$ is independent of $\text{X}_{\text{C}'}$ given $\text{X}_\text{C}$ in $\text{P}^*$). Since the conditional distribution of $Y$ given $X_C$ features is the same across $P$ and $P^*$, and the true labelling function $\text{f} \in \mathcal{H_C}$, the loss-minimizing models should also be the same. Defining $\text{h}_{\text{c,P}}^{\text{OPT}} = \arg\min_{\text{h}_\text{c} \in \mathcal{H}_c} \mathcal{L}_\text{P}(\text{h}_\text{c}, \text{f})$ and $\text{h}_{\text{c,P}^*}^{\text{OPT}} = \arg\min_{\text{h}_\text{c} \in \mathcal{H}_c} \mathcal{L}_{\text{P}^*}(\text{h}_\text{c}, \text{f})$, we obtain,

$$\text{h}_{\text{c,P}}^{\text{OPT}} = \text{h}_{\text{c,P}^*}^{\text{OPT}} = \text{f} \qquad (23)$$

And therefore, the term $\mathcal{L}_{\text{P}^*}(\text{h}_{\text{c,P}}^{\text{OPT}}, \text{h}_{\text{c,P}^*}^{\text{OPT}}) = 0$. Also, $\mathcal{L}_\text{P}(\text{h}_{\text{c,P}}^{\text{OPT}}, \text{f}) = \mathcal{L}_{\text{P}^*}(\text{h}_{\text{c,P}^*}^{\text{OPT}}, \text{f}) = 0$.

**Associational Model.** In contrast, an associational model may use a subset of $\text{X}$, $\text{X}_\text{S} \subseteq \text{X}$, that may not include all variables in the Markov Blanket, or may include the Markov Blanket but also include other extraneous variables. Let the set of such models be $\mathcal{H}_{AS}$. Note that we only consider $\mathcal{H}_{\text{AS}}$ to be the set of models for which $\mathcal{L}_\text{P}(\text{h}_{\text{as,P}}^{\text{OPT}}, \text{f}) = \mathcal{L}_{\text{P}^*}(\text{h}_{\text{as,P}^*}^{\text{OPT}}, \text{f}) = 0$ (otherwise, the loss bound for a model from $\mathcal{H}_{\text{AS}}$ is trivially larger). This can happen when a feature $x_i \notin X_C$ does not belong to the Markov Blanket under the causal graph, but belongs to the associational Markov Blanket under the probability distribution over $P$ or $P^*$.

Then, by definition of $f$, $f \notin \mathcal{H}_{\mathcal{AS}}$. Thus, $\text{h}_{\text{as,P}}^{\text{OPT}} \neq \text{f} \neq \text{h}_{\text{as,P}^*}^{\text{OPT}}$. Further, the loss-minimizing hypotheses $\text{h}_{\text{as,P}}^{\text{OPT}}$ and $\text{h}_{\text{as,P}^*}^{\text{OPT}}$ depend on the different distributions $\text{P}(\text{Y}|\text{X}_\text{S} = \text{x}_\text{s})$ and $\text{P}^*(\text{Y}|\text{X}_\text{S} = \text{x}_\text{s})$ and thus not necessarily equal. Hence, $\text{h}_{\text{a,P}}^{\text{OPT}} \neq \text{h}_{\text{a,P}^*}^{\text{OPT}}$.

For the other subset of associational models, since $\text{X}_\text{C} \subseteq \text{X}$, it is possible that an associational model includes only the causal features of $\text{Y}$. Therefore, in general, $\mathcal{L}_\text{P}(\text{h}_{\text{a,P}}^{\text{OPT}}, \text{h}_{\text{a,P}^*}^{\text{OPT}}) \geq 0$.

**Loss Bounds.** Hence, using Eq. 23, we write equation 19 for a causal model as:

$$\text{ODE}_{\text{P,P}^*}(\text{h}_\text{c}, \text{f}) = \mathcal{L}_{\text{P}^*}(\text{h}_\text{c}, \text{f}) - \mathcal{L}_{\text{S} \sim \text{P}}(\text{h}_\text{c}, \text{f}) \leq \text{disc}_{\text{L},\mathcal{H}_c}(\text{P}, \text{P}^*) + \text{IDE}_\text{P}(\text{h}_\text{c}, \text{f}) \qquad (24)$$

Using Eqns. 20 and 21, we obtain, with probability at least $1 - \delta$:

$$\text{ODE}_{\text{P,P}^*}(\text{h}_\text{c}, \text{f}) \leq \text{disc}_{\text{L},\mathcal{H}_c}(\text{P}, \text{P}^*) + \text{IDE-Bound}_\text{P}(\mathcal{H_C}, \mathcal{S}; \delta) \qquad (25)$$

$$\leq \text{disc}_{\text{L},\mathcal{H}_c}(\text{P}, \text{P}^*) + \text{IDE-Bound}_\text{P}(\mathcal{H_A}, \mathcal{S}; \delta) \qquad (26)$$

$$= \text{ODE-Bound}_{\text{P,P}^*}(\text{h}_\text{c}, \text{f}; \delta) \qquad (27)$$

Similarly, for an associational model, we obtain, with probability at least $1 - \delta$:

$$
\begin{aligned}
\texttt{ODE}_{P,P^*}(h_a, f) &= \mathcal{L}_{P^*}(h_a, f) - \mathcal{L}_{S \sim P}(h_a, f) \\
&\le \texttt{disc}_{L, \mathcal{H}_A}(P, P^*) + \texttt{IDE-Bound}_P(\mathcal{H}_A, \mathcal{S}; \delta) + \mathcal{L}_{P^*}(h_{a,P}^{\texttt{OPT}}, h_{a,P^*}^{\texttt{OPT}}) \quad (28) \\
&= \texttt{ODE-Bound}_{P,P^*}(h_a, f; \delta)
\end{aligned}
$$

Finally, using Definition 3 for discrepancy distance, we can state $\mathcal{H}_C \subseteq \mathcal{H}_A \Rightarrow \texttt{disc}_{L, \mathcal{H}_C}(P, P^*) \le \texttt{disc}_{L, \mathcal{H}_A}(P, P^*)$. Therefore, from Eq. 24 and 28, we claim with probability $(1 - \delta)^2$,

$$
\texttt{ODE-Bound}_{P,P^*}(h_c, f; \delta) \le \texttt{ODE-Bound}_{P,P^*}(h_a, f; \delta) \quad (29)
$$

$\square$

### A.2 GENERALIZATION OVER A SINGLE DATAPOINT

**Corollary 1.** *Consider a causal model* $h_c : X_C \to Y$ *and an associational model* $h_a : X \to Y$ *trained on a dataset* $S \sim P$. *Let* $(x, y) \in S$ *and* $(x', y') \notin S$ *be two input instances such that they share the same true labelling function* $y = f(x_c)$ *and* $y' = f(x'_c)$. *Then, the worst-case generalization error for a causal model on any such* $x'$ *is less than or equal to that for an associational model.* [Proof in Appendix Section A.2]

$$
\max_{x \in S, x' : y' = f(x'_c)} L_{x'}(h_c, f) - L_x(h_c, f) \le \max_{x \in S, x' : y' = f(x'_c)} L_{x'}(h_a, f) - L_x(h_a, f) \quad (5)
$$

*Proof.* Using the triangle inequality for the loss function, we obtain,

$$
L_{x'}(h, f) \le L_{x'}(h, h_P^{\texttt{OPT}}) + L_{x'}(h_P^{\texttt{OPT}}, h_{P^*}^{\texttt{OPT}}) + L_{x'}(h_{P^*}^{\texttt{OPT}}, f) \quad (30)
$$

$$
-L_x(h, f) \le -L_x(h, h_P^{\texttt{OPT}}) + L_x(h_P^{\texttt{OPT}}, f) \quad (31)
$$

Combining the two inequalities,

$$
\begin{aligned}
L_{x'}(h, f) - L_x(h, f) &\le L_{x'}(h, h_P^{\texttt{OPT}}) + L_{x'}(h_P^{\texttt{OPT}}, h_{P^*}^{\texttt{OPT}}) + L_{x'}(h_{P^*}^{\texttt{OPT}}, f) \\
&\quad - L_x(h, h_P^{\texttt{OPT}}) + L_x(h_P^{\texttt{OPT}}, f) \quad (32) \\
&\le \texttt{dist}_{\mathcal{H}}(x, x') + L_{x'}(h_{P^*}^{\texttt{OPT}}, f) + L_x(h_P^{\texttt{OPT}}, f) + L_{x'}(h_P^{\texttt{OPT}}, h_{P^*}^{\texttt{OPT}})
\end{aligned}
$$

where $\texttt{dist}(x, x') = \max_{h, h' \in \mathcal{H}} |L_{x'}(h, h') - L_x(h, h')|$, analogous to the $\texttt{disc}_L$ for distributions.

For a causal model, we know that $h_{c,P}^{\texttt{OPT}} = h_{c,P^*}^{\texttt{OPT}}$. Further, from Theorem 1 proof, $L_x(h_{c,P}^{\texttt{OPT}}, f) = L_x(h_{a,P}^{\texttt{OPT}}, f) = 0$ and same for $P^*$ and loss on $x'$.

Hence,

$$
L_{x'}(h_c, f) - L_x(h_c, f) \le \texttt{dist}_{\mathcal{H}_C}(x, x') \quad (33)
$$

$$
L_{x'}(h_a, f) - L_x(h_a, f) \le \texttt{dist}_{\mathcal{H}_A}(x, x') + L_{x'}(h_{a,P}^{\texttt{OPT}}, h_{a,P^*}^{\texttt{OPT}}) \quad (34)
$$

where $\texttt{dist}_{\mathcal{H}_C}(x, x') \le \texttt{dist}_{\mathcal{H}_A}(x, x')$ since $\mathcal{H}_C \subseteq \mathcal{H}_A$.

Next, we show that these bounds are tight. That is, there exists an associational model $h_a$ whose generalization error on $x'$ is exactly the RHS on Eqn. 34 and thus higher than the bound for any causal model. Below we prove by constructing one such $h_a$.

For simplicity in construction, let us select $H_C = \{h_{c,P}^{\texttt{OPT}}, h_{c,P^*}^{\texttt{OPT}}\}$ and $H_A = \{h_{c,P}^{\texttt{OPT}}, h_{c,P^*}^{\texttt{OPT}}, h_{a,P}^{\texttt{OPT}}, h_{a,P^*}^{\texttt{OPT}}\}$. Thus $\texttt{dist}_{H_C}(x, x') = 0$ whereas $\texttt{dist}_{H_A}(x, x') \ge 0$. We obtain the ODE-generalization bound for a causal model,

$$
L_{x'}(h_c, f) - L_x(h_c, f) \le L_{x'}(h_{a,P^*}^{\texttt{OPT}}, f) + L_x(h_{a,P}^{\texttt{OPT}}, f) = \texttt{ODE-Bound}_{x,x'}(h_c, f; H_C) \quad (35)
$$

Let us now construct an associational model, $h_a^{\dagger}$, such that:

$$
\begin{aligned}
L_{x'}(h_a^{\dagger}, f) &= L_{x'}(h_a^{\dagger}, h_P^{\texttt{OPT}}) + L_{x'}(h_P^{\texttt{OPT}}, h_{P^*}^{\texttt{OPT}}) + L_{x'}(h_{P^*}^{\texttt{OPT}}, f) \\
-L_x(h_a^{\dagger}, f) &= -L_x(h_a^{\dagger}, h_P^{\texttt{OPT}}) + L_x(h_P^{\texttt{OPT}}, f)
\end{aligned} \quad (36)
$$

A simple construction for the above equalities is to select $P$ and $P^*$, and correspondingly $\mathtt{x}^\dagger$ and $\mathtt{x}'^\dagger$ such that $\mathtt{L}_{\mathtt{x}^\dagger}(\mathtt{h}^{\mathtt{OPT}}_{\mathtt{a},\mathtt{P}},\mathtt{f}) = \mathtt{L}_{\mathtt{x}'^\dagger}(\mathtt{h}^{\mathtt{OPT}}_{\mathtt{a},\mathtt{P}^*},\mathtt{f}) = 0$ and $\mathtt{L}_{\mathtt{x}'^\dagger}(\mathtt{h}^{\mathtt{OPT}}_{\mathtt{a},\mathtt{P}},\mathtt{h}^{\mathtt{OPT}}_{\mathtt{a},\mathtt{P}^*}) > 0$. Further, $\mathtt{h}^\dagger_{\mathtt{a}}$ can be selected such that $\mathtt{L}_{\mathtt{x}'^\dagger}(\mathtt{h}^\dagger_{\mathtt{a}},\mathtt{h}^{\mathtt{OPT}}_{\mathtt{P}}) = 0$ and $\mathtt{L}_{\mathtt{x}^\dagger}(\mathtt{h}^\dagger_{\mathtt{a}},\mathtt{h}^{\mathtt{OPT}}_{\mathtt{P}}) = 0$.

Then, using Eqn. 36,

$$
\begin{aligned}
\mathtt{L}_{\mathtt{x}'^\dagger}(\mathtt{h}^\dagger_{\mathtt{a}},\mathtt{f}) - \mathtt{L}_{\mathtt{x}^\dagger}(\mathtt{h}^\dagger_{\mathtt{a}},\mathtt{f}) &= \mathtt{L}_{\mathtt{x}'}(\mathtt{h}^\dagger_{\mathtt{a}},\mathtt{h}^{\mathtt{OPT}}_{\mathtt{P}}) - \mathtt{L}_{\mathtt{x}}(\mathtt{h}^\dagger_{\mathtt{a}},\mathtt{h}^{\mathtt{OPT}}_{\mathtt{P}}) + \mathtt{L}_{\mathtt{x}'^\dagger}(\mathtt{h}^{\mathtt{OPT}}_{\mathtt{a},\mathtt{P}^*},\mathtt{f}) + \mathtt{L}_{\mathtt{x}^\dagger}(\mathtt{h}^{\mathtt{OPT}}_{\mathtt{a},\mathtt{P}},\mathtt{f}) + \mathtt{L}_{\mathtt{x}'^\dagger}(\mathtt{h}^{\mathtt{OPT}}_{\mathtt{a},\mathtt{P}},\mathtt{h}^{\mathtt{OPT}}_{\mathtt{a},\mathtt{P}^*}) \\
&= 0 + \mathtt{L}_{\mathtt{x}'^\dagger}(\mathtt{h}^{\mathtt{OPT}}_{\mathtt{a},\mathtt{P}^*},\mathtt{f}) + \mathtt{L}_{\mathtt{x}^\dagger}(\mathtt{h}^{\mathtt{OPT}}_{\mathtt{a},\mathtt{P}},\mathtt{f}) + \mathtt{L}_{\mathtt{x}'^\dagger}(\mathtt{h}^{\mathtt{OPT}}_{\mathtt{a},\mathtt{P}},\mathtt{h}^{\mathtt{OPT}}_{\mathtt{a},\mathtt{P}^*}) \\
&> \mathtt{L}_{\mathtt{x}'^\dagger}(\mathtt{h}^{\mathtt{OPT}}_{\mathtt{a},\mathtt{P}^*},\mathtt{f}) + \mathtt{L}_{\mathtt{x}^\dagger}(\mathtt{h}^{\mathtt{OPT}}_{\mathtt{a},\mathtt{P}},\mathtt{f}) \\
&= \mathtt{ODE\text{-}Bound}_{\mathtt{x},\mathtt{x}'}(\mathtt{h}_{\mathtt{c}},\mathtt{f};\mathtt{H}_{\mathtt{C}})
\end{aligned}
$$
(37)

where the last equality comes from Eqn. 35. Combining Eqns. 37 and 33, we obtain,

$$
\max_{\mathtt{x},\mathtt{x}'} \mathtt{L}_{\mathtt{x}'}(\mathtt{h}_{\mathtt{c}},\mathtt{f}) - \mathtt{L}_{\mathtt{x}}(\mathtt{h}_{\mathtt{c}},\mathtt{f}) \le \max_{\mathtt{x},\mathtt{x}'} \mathtt{ODE\text{-}Bound}_{\mathtt{x},\mathtt{x}'}(\mathtt{h}_{\mathtt{c}},\mathtt{f};\mathtt{H}_{\mathtt{C}}) \le \max_{\mathtt{x},\mathtt{x}'} \mathtt{L}_{\mathtt{x}'}(\mathtt{h}_{\mathtt{a}},\mathtt{f}) - \mathtt{L}_{\mathtt{x}}(\mathtt{h}_{\mathtt{a}},\mathtt{f})
$$
(38)

where the maximum is over $\mathtt{x} \in \mathtt{S}$ and $\mathtt{x}' \notin \mathtt{S}$ such that $\mathtt{y}' = \mathtt{f}(\mathtt{x}'_{\mathtt{c}})$. $\qquad\square$

# B  DIFFERENTIAL PRIVACY GUARANTEES WITH TIGHTER DATA-DEPENDENT BOUNDS

In this section we provide the differential privacy guarantee of a causal model based on a recent method (Papernot et al., 2017) that provides tighter data-dependent bounds. We first present a Lemma on generalization of causal models and then prove the main result.

As a consequence of Theorem 1, another generalization property of causal models is that the models trained on data from two different distributions $P(X)$ and $P^*(X)$ are likely to output the same value for a new input.

**Lemma 3.** *Under the conditions of Theorem 1, let $h_{c,P}$ be a causal model trained on distribution $P$ and let $h_{c,P^*}$ be a model trained on $P^*$. Similarly, let $h_{a,P}$ and $h_{a,P^*}$ be correlational models trained on $P$ and $P^*$ respectively. Assume that the correlational and causal models report the same accuracy $\alpha$. Then for any new data input $x$,*

$$
\Pr(h_{c,P}(x) = h_{c,P^*}(x)) \ge \Pr(h_{a,P}(x) = h_{a,P^*}(x))
$$

*As the size of the training sample $N \to \infty$, the LHS$\to 1$.*

*Proof.* Let $h^{OPT}_{a,P} = \arg\min_{h \in \mathcal{H}_A} \mathcal{L}_P(h,f)$ and similarly let $h^{OPT}_{a,P^*} = \arg\min_{h \in \mathcal{H}_A} \mathcal{L}_{P*}(h,f)$ be the loss-minimizing hypotheses under these two distributions, where $\mathcal{H}$ is the set of hypotheses. We can analogously define $h^{OPT}_{c,P}$ and $h^{OPT}_{c,P^*}$. For a causal model, we know from Theorem 1 that $h^{OPT}_{c,P} = h^{OPT}_{c,P^*}$. As $N \to \infty$, each of models on $P$ and $P^*$ approach their loss-minimizing functions. Then, for any input $x$,

$$
L(h_{c,P}(x), h_{c,P^*}(x)) = L(h^{OPT}_{c,P}(x), h^{OPT}_{c,P^*}(x)) = 0
$$
(39)

$$
L(h_{a,P}(x), h_{a,P^*}(x)) = L(h^{OPT}_{a,P}(x), h^{OPT}_{a,P^*}(x)) \ge 0
$$
(40)

$$
\Rightarrow \Pr(h_{c,P}(x) = h_{c,P^*}(x)) = 1 \ge \Pr(h_{a,P}(x) = h_{a,P^*}(x))
$$
(41)

Under finite samples, let the accuracy of each model be $\alpha$. Then for any new input, the causal models predict the same output if they both match $h^{min}_{c,P} = h^{min}_{c,P^*}$ or both do not match. The probability of this event can be captured by the accuracy of the models: $\alpha^2$ if both models match $h^{min}_{c,P}$, and $(1-\alpha)^2$ if they do not.

$$
\Pr(h_{c,P} = h_{c,P^*}) = \alpha^2 + (1-\alpha)^2
$$
(42)

For correlational models, $h^{OPT}_{a,P} \ne h^{OPT}_{a,P^*}$ for some $x$. For those $x$, assuming the same accuracy $\alpha$,

$$
\Pr(h1 = h2) = 2\alpha(1-\alpha)
$$
(43)

Since $2\alpha(1-\alpha) \le \alpha^2 + (1-\alpha)^2 \ \forall \alpha \in [0,1]$, we have

$$
\Pr(h_{c,P} = h_{c,P^*}) \ge \Pr(h_{a,P} = h_{a,P^*})
$$

$\qquad\square$

Based on the above generalization property, we now show that causal models provide stronger differential privacy guarantees than corresponding associational models. We utilize the subsample and aggregate technique (Dwork et al., 2014) that was extended for machine learning in Hamm et al. (2016) and Papernot et al. (2017), for constructing a differentiably private model. The framework considers $M$ arbitrary teacher models that are trained on a separate subsample of the dataset without replacement. Then, a student model is trained on some auxiliary unlabeled data with the (pseudo) labels generated from a majority vote of the teachers. Differential privacy can be achieved by either perturbing the number of votes for each class (Papernot et al., 2017), or perturbing the learnt parameters of the student model (Hamm et al., 2016). For any new input, the output of the model is a majority vote on the predicted labels from the M models. The privacy guarantees are better if a larger number of teacher models agree on each input, since by definition the majority decision could not have been changed by modifying a single data point (or a single teacher's vote). Since causal models generalize to new distributions, intuitively we expect causal models trained on separate samples to agree more. Below we show that for a fixed amount of noise, a causal model is $\epsilon_c$-DP compared to $\epsilon$-DP for a associational model, where $\epsilon_c \leq \epsilon$.

**Theorem 3.** *Let* $D$ *be a dataset generated from possibly a mixture of different distributions* $P(X, Y)$ *such that* $P(Y|X_C)$ *remains the same. Let* $n_j$ *be the votes for the jth class from M teacher models. Let* $\mathcal{M}$ *be the mechanism that produces a noisy max,* $\arg\max_j\{n_j + Lap(2/\gamma)\}$. *Then the privacy budget* $\epsilon$ *for a causal model is lower than that for the associational model with the same accuracy.*

*Proof.* Consider a change in a single input example $(x, y)$, leading to a new $D'$ dataset. Since sub-datasets are sampled without replacement, only a single teacher model can change in $D'$. Let $n'_j$ be the vote counts for each class under $D'$. Because the change in a single input can only affect one model's vote, $|n_j - n'_j| \leq 1$.

Let the noise added to each class be $r_j \sim Lap(2/\gamma)$. Let the majority class (class with the highest votes) using data from $D$ be $i$ and the class with the second largest votes be $j$. Let us consider the minimum noise $r^*$ required for class $i$ to be the majority output under $\mathcal{M}$ over $D$. Then,

$$n_i + r^* > n_j + r_j$$

For $i$ to have the maximum votes using $\mathcal{M}$ over $D'$ too, we need,

$$n'_i + r_i > n'_j + r_j$$

In the worst case, $n'_i = n_i - 1$ and $n'_j = n_j + 1$ for some $j$. Thus, we need,

$$n_i - 1 + r_i > n_j + 1 + r_j \Rightarrow n_i + r_i > n_j + 2 + r_j \qquad (44)$$

which shows that $r_i > r* + 2$. Note that $r* > r_j - (n_i - n_j)$. We have two cases:

**CASE I:** The noise $r_j < n_i - n_j$, and therefore $r^* < 0$. Writing $\Pr(i|D')$ to denote the probability that class $i$ is chosen as the majority class under $D'$,

$$P(i|D') = P(r_i \geq r^* + 2) \qquad\qquad = 1 - 0.5\exp(\gamma)\exp(\tfrac{1}{2}\gamma r^*) \qquad (45)$$

$$= 1 - \exp(\gamma)(1 - P(r_i \geq r^*)) \qquad\qquad = 1 - \exp(\gamma)(1 - P(i|D)) \qquad (46)$$

where the equations on the right are due to Laplace c.d.f. Using the above equation, we can write:

$$\frac{P(i|D')}{P(i|D)} = \exp(\gamma) + \frac{1 - \exp(\gamma)}{P(i|D)} = \exp(\gamma) + \frac{1 - \exp(\gamma)}{P(r_i \geq r^*)} \leq \exp(\epsilon) \qquad (47)$$

for some $\epsilon > 0$. As $P(i|D) = P(r_i \geq r^*)$ increases, the ratio decreases and thus the effective privacy budget ($\epsilon$) decreases. Thus, a model with a lower $r^*$ (effectively higher $|r^*|$) will exhibit the lowest $\epsilon$.

Below, we show that $|r^*|$ is higher for a causal model, and thus $P(r_i \geq r^*)$ is higher. Intuitively, $|r^*|$ is higher when there is more consensus between the M teacher models since $|r^*|$ is the difference between the votes for the highest voted class with the votes for the second-highest class.

Let us consider two causal teacher models $h1_c$ and $h2_c$, and two associational teacher models, $h1$ and $h2$. From Lemma 3, for any new $x$, and for same accuracies of the models, there is more consensus among causal models.

$$P(h1_c(x) = h2_c(x)) \geq P(h1(x) = h2(x)) \qquad (48)$$

Hence $r_c^* \leq r^*$. From Equation 47, $\epsilon_c \leq \epsilon$.

**CASE II:** The noise $r_j >= n_i - n_j$, and therefore $r^* >= 0$. Following the steps above, we obtain:

$$P(i|D') = P(r_i \geq r^* + 2) \qquad\qquad = 0.5 \exp(-\gamma) \exp(-\frac{1}{2}\gamma r^*) \tag{49}$$

$$= \exp(-\gamma)(P(r_i \geq r^*)) \qquad\qquad = \exp(-\gamma)(P(i|D)) \tag{50}$$

Thus, the ratio does not depend on $r^*$.

$$\frac{P(i|D')}{P(i|D)} = \exp(-\gamma) \tag{51}$$

Under CASE II when the noise is higher to the differences in votes between the highest and second highest voted class, causal models provide the same privacy budget as associational models.

Thus, overall, $\epsilon_c \leq \epsilon$. $\qquad\qquad\qquad\qquad\qquad\qquad\qquad\qquad\qquad\qquad\qquad\qquad\qquad\square$

## C  INFINITE SAMPLE ROBUSTNESS TO MEMBERSHIP INFERENCE ATTACKS

**Corollary 2.** *Let* $h_{c,S}^{min}$ *be a causal model trained using empirical risk minimization on a dataset* $S \sim P(X, Y)$ *with sample size* $N$. *As* $N \to \infty$, *membership advantage* $Adv(\mathcal{A}, h_{c,S}^{min}) \to 0$.

*Proof.* $h_{c,S}^{min}$ can be obtained by empirical risk minimization.

$$h_{c,S}^{min} = \arg\min_{h \in \mathcal{H}_c} \mathcal{L}_{S \sim P}(h, f) = \arg\min_{h \in \mathcal{H}_c} \frac{1}{N} \sum_{i=1}^{N} L_{x_i}(h, f) \tag{52}$$

As $|S| = N \to \infty$, $h_{c,S}^{min} \to h_{c,P}^{OPT}$. Suppose now that there exists another $S'$ of the same size such that $S' \sim P^*$. Then as $|S'| \to \infty$, $h_{c,S'}^{min} \to h_{c,P^*}^{OPT}$.

From Theorem 1, $h_{c,P}^{OPT} = h_{c,P^*}^{OPT}$. Thus,

$$\lim_{N\to\infty} h_{c,S}^{min} = \lim_{N\to\infty} h_{c,S'}^{min} \tag{53}$$

Equation 53 implies that as $N \to \infty$, the learnt $h_{c,S}^{min}$ does not depend on the training set, as long as the training set is sampled from any distribution $P^*$ such that $P(Y|X_C) = P^*(Y|X_C)$. That is, being the global minimizer over distributions, $h_{c,S}^{min} = h_{c,P}^{OPT}$ does not depend on its training set. Therefore, $h_{c,S}^{min}(x)$ is independent of whether $x$ is in the training set.

$$\lim_{N\to\infty} Adv(\mathcal{A}, h_{c,S}^{min}) = Pr(\mathcal{A} = 1|b = 1) - Pr(\mathcal{A} = 1|b = 0)$$

$$= \mathbb{E}[\mathcal{A}|b = 1] - \mathbb{E}[\mathcal{A}|b = 0] = E[\mathcal{A}(h_{c,S}^{min})|b = 1] - \mathbb{E}[\mathcal{A}(h_{c,S}^{min})|b = 0] \tag{54}$$

$$= \mathbb{E}[\mathcal{A}(h_{c,S}^{min})] - \mathbb{E}[\mathcal{A}(h_{c,S}^{min})] = 0$$

where the second last equality follows since any function of $h_{c,S}^{min}$ is independent of the training dataset. $\qquad\qquad\qquad\qquad\qquad\qquad\qquad\qquad\qquad\qquad\qquad\qquad\qquad\qquad\qquad\square$

## D  ROBUSTNESS TO ATTRIBUTE INFERENCE ATTACKS

In addition to revealing membership in the training set, a model may also reveal the value of individual sensitive features of a test input, given partial knowledge of its features. For instance, given a training dataset of HIV patients, an adversary may infer other attributes of a person (e.g., genetic information) given that they know their demographics and other public features. As another example, it can be possible to infer a person's face based on hill climb on the output score for a face detection model (Fredrikson et al., 2015). Model inversion is not always due to a fault in learning: a model may learn a true, generalizable relationship between features and the outcome, but still be vulnerable to a model inversion attack. This is because given (k-1) features and the true outcome label, it is possible to guess the kth feature by brute-force search on output scores generated by the model. However, inversion based on learning correlations between features, e.g., using some demographics to predict disease, can be alleviated by causal models, since a feature will not be included in a model unless it directly affects the outcome.

**Definition 7** (From Yeom et al. (2018)). *Let h be a model trained on a dataset $\mathcal{D}(X,Y)$. Let A be an adversary with access to h, and a partial test input $x_A \subset x$. The attribute advantage of the adversary is the difference between true and false positive rates in guessing the value of a sensitive feature $\mathtt{x_s} \notin \mathtt{x_A}$. For a binary $\mathtt{x_s}$,*

$$\mathtt{Adv}(\mathcal{A}, \mathtt{h}) = \Pr(\mathcal{A} = 1|\mathtt{x_s} = 1) - \Pr(\mathcal{A} = 1|\mathtt{x_s} = 0) \tag{55}$$

**Theorem 5.** *Given a dataset $\mathcal{D}(X, Y)$ of size $N$ and a structural causal network that connects $\mathtt{X}$ to $\mathtt{Y}$, a causal model $\mathtt{h_c}$ makes it impossible to infer non-causal features.*

*Proof.* The proof follows trivially from definition of a causal model. $\mathtt{h_c}$ includes only causal features during training. Thus, $\mathtt{h(x)}$ is independent of all features not in $\mathtt{X_c}$.

$$
\begin{aligned}
\mathtt{Adv}(\mathcal{A}, \mathtt{h}) &= \Pr(\mathcal{A} = 1|\mathtt{x_s} = 1) - \Pr(\mathcal{A} = 1|\mathtt{x_s} = 0) \\
&= \Pr(\mathcal{A}(\mathtt{h}) = 1|\mathtt{x_s} = 1) - \Pr(\mathcal{A}(\mathtt{h}) = 1|\mathtt{x_s} = 0) = \Pr(\mathcal{A}(\mathtt{h}) = 1) - \Pr(\mathcal{A}(\mathtt{h}) = 1) = 0
\end{aligned} \tag{56}
$$

$\square$

# E EXPERIMENTS

## E.1 DATASET DISTRIBUTION

The target model is trained using the synthetic training and test data generated using the bnlearn library. We first divide the total dataset into training and test dataset in a 60:40 ratio. Further, the output of the trained model for each of the training and test dataset is again divided into 50:50 ratio. The training set for the attacker model consists of confidence values of the target model for the training as well as the test dataset. The relation is explained in Figure 4. Note that the attacker model is trained on the confidence output of the target models.

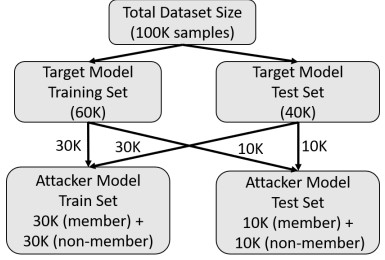

Figure 4: Dataset division for training target and attacker models.

## E.2 FINE-GRAINED ATTACK ANALYSIS OF LEARNED CAUSAL MODELS.

We expand the evaluation of the attack accuracy for a learnt causal model on the `Sachs` dataset and `Akt` outcome. We increment the non-causal features to the learned model in addition to the true causal features. Table 2 shows the attack and prediction accuracy for this model when trained with the true causal, learned causal model with 1 and 2 non-causal features, and the results for the corresponding DNN model.

| Acc. (%) | True Model | Learned Causal (2 causal +) | | DNN |
|---|---|---|---|---|
| | 2 causal parents | 1 non-causal parent | 2 non-causal parents | |
| Attack | **50** | **52** | **61** | **76** |
| Pred. | 79 | 75 | 68.8 | 73 |

Table 2: *Attack and Prediction accuracy comparison across models for* `Sachs` *dataset and* `Akt` *output variable.*

