# OpenReview forum: "Alleviating Privacy Attacks via Causal Learning"
_ICLR.cc/2020/Conference — Reject_

### Official Review · AnonReviewer3 · 2019-10-21
**Official Blind Review #3**

**Rating:** 6

**Review:**

Overview: This paper discusses the risk of membership inference attacks that deep neural networks might face when used in a practical manner on real world datasets. Membership inference attacks can result in privacy breaches, a significant concern for many fields who might stand to benefit from using deep learning in applications. The authors demonstrate how attack accuracy goes up when one dataset is used for training while another altogether is used for testing. They propose the use of causal learning approaches in order to negate risk of membership inference attacks. Causal models can handle distribution shifts across datasets because they learn using a causal structure.

Contributions: In the theory part of the paper, the authors provide several proofs demonstrating that causal models have stronger differential privacy guarantees than association modes, that causal models trained on large samples are able to protect the dataset against attacks, and that causal models trained on smaller samples still have higher protection than association models trained on similarly sized samples. In addition to theoretical contributions, the authors also provide an experimental evaluation using 4 accepted experimental datasets.


Questions and Comments:
Page 2: “...while association modes exhibit upto 80%...” -> “...up to...”
My expertise is not in causal learning or structures, so I have a few questions about using it in practice. You mentioned that the datasets used in the experimental section were used in order to avoid errors in learning causal structure.
How likely is it to have these errors using a different dataset?
How long/how much effort does it take to figure out conditional probability table? Is this a significant amount of time compared to training? Is it automatic or manually done by humans?
If it is done by humans, is it plausible to assume that every dataset implicitly contains a causal structure (not including random walks)?
Your experimental results suggest that the causal model can learn on smaller amounts of data than the DNN. Does this scale for even larger input parameter datasets as well, such as water?
You mention this potentially being used to prevent attacks on real-world applications, such as HIV patient prediction/classification systems. Do you believe that your results prove causal models will scale to datasets that contain these kinds of complex causal structures?
Could you provide the layer architectures of all three models used for your experiments? Are these out-of-the-box solutions from libraries, or something more custom built?

How would the causal model perform compared to state of the art techniques for these datasets, in both accuracy and attack protection? I understand that isn't the main point of this paper, this is me being curious.

I give this paper a borderline acceptance, based upon the fact that the above questions need to be addressed. I'm not sure its clear to see how the experimental results demonstrate that the causal model definitely outperforms DNNs in all cases. I would like to hear the author's defense of the method when it comes to datasets with higher numbers of features, specifically the water dataset.


**Experience Assessment:**

I do not know much about this area.

**Review Assessment: Checking Correctness Of Derivations And Theory:**

I did not assess the derivations or theory.

**Review Assessment: Checking Correctness Of Experiments:**

I assessed the sensibility of the experiments.

**Review Assessment: Thoroughness In Paper Reading:**

I read the paper at least twice and used my best judgement in assessing the paper.

---

> ### Author Response · Authors · 2019-11-11
> **Response to Review 3**
>
> > 1.How likely is it to have errors in learning causal structure using a different dataset?
>
> Learning the correct causal graph from a single dataset is still an open problem, so we think it is likely that there will be errors using a different dataset. The Bnlearn library provide a few functions for learning causal structure from a given dataset. We tested a simple hill-climbing algorithm and observed it to return the true causal parents for each of the output variable in our benchmark datasets. Hence, existing methods might be useful in learning causal structures. In addition, several recent research work propose techniques to learn causal models without having to learn the correct causal graph [Arjovsky et al., Ke et al.]. Our work provides a theoretical basis for these techniques to alleviate membership inference attacks.
> [Arjovsky et al.] -- Invariant Risk Minimization
> [Ke et al.] -- Learning Neural Causal Models From Unknown Interventions
>
> > 2. How long/how much effort does it take to figure out conditional probability table?
>
> Given the structure of the causal graph, it is easy to compute the conditional probability table. In fact, since each variable's probability distribution is conditioned only on its parents and is independent of others' conditional probability distributions, we can find the correct parameters for each one independently using simple maximum likelihood.  The dataset itself does not contain the causal structure, but the data distribution may have been derived from the structure of the causal graph.
>
> > 3. Your experimental results suggest that the causal model can learn on smaller amounts of data than the DNN. Does this scale for even larger input parameter datasets?
>
> We haven't evaluated on larger datasets. But in principle, since a causal model uses fewer features than an associational model like DNN, it should learn better with fewer number of samples.
>
> > 4. Do you believe that your results prove causal models will scale to datasets that contain these kinds of complex causal structures?
>
> We provided the HIV example as a motivating example, and showed a theoretical basis on why it is important to make an effort to learn causal models. However, our work does not focus on the empirical learning of a causal model. While there are some recent works (see our references to Arjovksy et al. and Ke et al.), we believe the field needs to do more work before we can think of implementing causal models in complicated causality settings like health.
>
> > 5. Could you provide the layer architectures of all three models used for your experiments?
>
> The associational model is a multilayer perceptron (MLP) with 3 hidden layers of 128, 512 and 128 nodes respectively. The learning rate is set to 0.0001 and the model is trained for 10000 steps. The attacker model has 2 hidden layers with 5 nodes each, a learning rate of 0.001, and is trained for 5000 steps. Both models use Adam optimizer, ReLU for the activation function, and cross entropy as the loss function.  We chose these parameters to ensure model convergence.
>  For the causal models, we use the maximum likelihood estimation method to learn the probability table of the model for the output variable using the known causal structure.
>
> > 6. How would the causal model perform compared to state of the art techniques for these datasets, in both accuracy and attack protection?
>
> Since we know the data-generating process for these datasets, the best possible accuracy on these datasets would be for a model that uses the true causal structure and the true conditional probability values. We implemented that method and found that the accuracies and attack performance are almost identical to the causal model we present in the paper.
>
> > 7. I'm not sure that the causal model definitely outperforms DNNs in all cases.
>
> If we understand correctly, the concern on outperforming is with respect to the privacy guarantees. Our experimental results show that for models with higher number of features, associational models tend to overfit on the distribution while causal models learn to predict only using the causal features. Specifically, figure 3b shows that the attack accuracy for DNNs on the water dataset (high number of features) is the highest as compared to all the other datasets, while the attack accuracy for the causal model is close to a random guess. Overall,  since causal models generalize well across distributions, membership inference attacks are harder to perform on these models as compared to associational model. That said, when trying to learn the causal structure from data (rather than using known causal structure), we did face difficulty with the Water dataset due to its extreme probabilities, but we emphasize that our main contribution is to show the connection between privacy and causal learning, both theoretically and empirically. We point the reader to related work that presents state-of-the-art methods on learning causal structure.

---

### Official Review · AnonReviewer1 · 2019-10-29
**Official Blind Review #1**

**Rating:** 3

**Review:**

The paper proposes using causal learning models for alleviating privacy attacks, i.e. membership inference attacks. The paper proves that causal models trained on sufficiently large samples are robust to membership inference attacks; they confirm the theories with experiments on 4 synthetic data.
The paper is well written; theoretical proof seems correct as it combines proof of differential privacy guarantees  in Papernot et al. 2017, robustness to membership attacks in Yeom et al. 2018 with the generalization property of causal models from Pearl 2009 and Peters et al. 2017. Results are presented clearly. The paper is novel as the authors claimed they provide the first analysis of privacy benefits of causal models.
The main concern of this paper is the results are only confirmed on synthetic data, where all the 4 datasets are generated from known Bayesian networks (i.e., causal graphs). It doesn’t matter if these Bayesian nets are complex or not, because most of the experiments are done with the known true causal models except the last experiment in Figure 3c. Even with learnt causal models, they were learning a Bayesian net from too optimistic data that were indeed generated from Bayesian nets, but these are usually not true for real world data. So evaluations on real dataset, or other synthetic data that are not generated from Bayesian nets are necessary for validating the methods.
Another question is about the ‘causal models are known to be invariant to the training distribution and hence generalize well to shifts between samples from the same distribution and across different distributions.’  More explanations about ‘invariance’ is needed. For example, in Figure 2a and Figure 3a, causal models have similar performance (except Alarm data) with DNN models on test 2, where test samples are generated from different distributions than training samples. Also in Figure 3b, the attack accuracy are no different between causal models and DNN on test 1.
The last minor question is why only parents of Y are included in causal models in the experiments, but not the Markov blanket as stated earlier in Figure 1.






**Experience Assessment:**

I have published one or two papers in this area.

**Review Assessment: Checking Correctness Of Derivations And Theory:**

I did not assess the derivations or theory.

**Review Assessment: Checking Correctness Of Experiments:**

I assessed the sensibility of the experiments.

**Review Assessment: Thoroughness In Paper Reading:**

I made a quick assessment of this paper.

---

> ### Author Response · Authors · 2019-11-11
> **Response to Review 1**
>
>
> 	> 1. The main concern of this paper is the results are only confirmed on synthetic data, where all the 4 datasets are generated from known Bayesian networks (i.e., causal graphs).
>
> Our main claim is that "true causal models have stronger differential privacy guarantees as compared to associational models". The key contribution of the paper is to show the privacy benefits of causal learning and motivate the adoption of these techniques for sensitive applications. For this, we want to empirically  show that these models are robust to membership inference attacks as compared to associational models.  Hence, we aim to evaluate on models where the causal structure is known apriori. Therefore, we chose to use the Bayesian networks and alter their probabilities to create distributions from different domain.
>
> 	> 2. More explanations about ‘invariance’ is needed.
>
> The statement on invariance in the paper assumes a true/ideal causal model, where we assume that the model class for training is expressive enough to capture P(Y|X_C) and that the dataset size is large enough to prevent any estimation errors. Under these conditions, as long as $P(Y|X_C) = P*(Y|X_C)$ remains invariant across any two distributions P and P*, then the error on a particular input, say $x_i$ is the same under P and P*. This notion of same error is captured by the "invariant" statement.  Note that this may not be true for an associational model, since it may have captured associations P(Y|X) that are not stable across distributions $P(Y|X)!= P*(Y|X)$.  We have clarified this in the revised version of the abstract. For completeness, we also mention that causal models are not invariant to covariate shifts in the features (changes in distribution of X or X_C), and thus the overall error on P* can be different from P.
>
> 	> 3. Causal models have similar performance (except Alarm data) with DNN models on test 2. The attack accuracy are no different between causal models and DNN on test 1.
>
> As you rightly point for Figures 2a and 3a, causal model have similar performance to DNNs on the test data. This is likely because the associational DNN models also were invariant in this case i.e., had similar error on train and test across distributions. Finally, to clarify, test 1 has the same distribution as the train dataset, so we expect to see 50% (random guess) attack accuracy for both causal and DNN models. The real benefit of the invariance property is that the attack accuracy continues to be 50% even when the test distribution is changed.
>
>
> 	> 4. Why only parents of Y are included in causal models in the experiments?
>
> This was just for convenience--wherever possible, we chose outcomes that had no descendants in the causal graph, so that the Markov Blanket includes only parents of Y. Further, the prediction function in the bnlearn library uses only parents and hence was the primary choice for implementation.

---

### Official Review · AnonReviewer2 · 2019-11-07
**Official Blind Review #2**

**Rating:** 3

**Review:**

Summary:

   The authors consider a transfer learning problem where the source distribution is P(X,Y) while the target distribution is P*(X,Y) and classifier is trained on data from the source distribution. They also assume that the causal graph generating the data (X and Y) is identical while the conditional probabilities (mechanisms) could change between the source and the target. Further, they assume that if X_C is the Markov Blanket for variable Y in P and P*,  then P(Y|X_C) = P*(Y|X_C). Therefore the best predictor in terms of cross entropy loss for both distributions is identical if it focuses on the variables in the Markov Blanket. Authors define "causal hypothesis" as the one that uses only variables in the Markov Blanket (X_C) to predict Y.

 In this setting, the authors show two sets of results: a) Out of Distribution Generalization Error is less for the optimal causal predictors from any class of causal hypotheses under any loss function. b) If we tweak the definition of differential privacy where neighboring datasets are defined by replacing one of the variables in the training set by a sample from the test distribution (I have lots of questions about this definition later on), then the authors show that causal classifiers that depend only on the Markov Blanket has lower sensitivity to the change than that of associative classifiers (that use all features). This is used to prove that the causal ones have tighter differential privacy guarantees than the associative ones
 c) Using the differential privacy results, they also show that optimal causal classifiers are more resistant to membership attacks.


The authors demonstrate the results through membership attack accuracies on causally trained and associative models on 4 datasets where the causal graph and the parameters (conditional probabilities) of the Bayesian Network are known apriori.


Major Issues:
    I have lots of issues with the theory in the paper. Thats the main reason for my recommendation.

  1. Why is h_{c,P}^{OPT} = h_{c,P*}^{OPT} ?? (Equation 23 in Page 11) - Authors say that since the markov blanket is the only thing used to predict Y for causal predictors and P (Y|X_C) = P *(Y|X_C), this should be true. But I have problems with this statement/argument. First of all, this is claimed for for any loss function L (not just Cross Entropy Loss) and particularly for a generic hypothesis class H_C (that depends only on Markov Blanket).

Consider the following counter example - Suppose all the features are in the Markov Blanket of Y (even a simpler case where all features are causal parents of Y). Suppose the true P (Y|X) is a logistic model with weights w_1 on one part of the domain D_1 and with weights w_2 for another part of the domain D_2. Suppose P is a mixture distribution of D_1 and D_2. P* is another mixture distribution (mixed differently) of D_1 and D_2.

Suppose I consider the class of logistic classifiers (but a single logistic model with one weight w) to be my hypothesis class and I use the standard logistic loss (logistic regression) on P, since a single logistic model with one slope cannot match the different slopes in different parts of the domain, it will result in some weight vector w^{opt}_1. Now, since the mixtures of D_1 and D_2 are different in the P* (but P (Y|X ) is identical), the optimal w^{opt}_2 for the P* will be different.

So for arbitrary hypothesis classes (that do not capture the true P(Y|X_c)) and for a non cross entropy loss - clearly this does not hold at all !! Covariate shifts amongst X_C alone with create a different classifier for an arbitrary loss (even if P (Y|X_C) is the same across both). In fact, the only way I see to salvage this is to assume Cross Entropy loss and talk about all soft classifiers (without restrictions to hypothesis class). But even if thats the case, then the best associational model will be the one that uses the Markov Blanket too ! .

This claim about h_C is crucially used in proof of theorem 1 (Page 11) and Proof of Corollary 1 (Page 13). This is fundamental to all theorems later. That brings into question the validity of many (if not all) the theoretical results in the paper. Authors must address this.

2. Issues regarding definition of certain quantities.

a) In equation 5 a quantity max_{x,x'} L_{x sampled from P} (h_c,f) - L{x' sampled from P*} (h_c,f) is defined - the inner quantity is a random quantity that depends on the samples x and x', Then what is the max operator over ?? - What does it mean to have worst case over samples from a distribution ??? Does it mean samples from two different domains ??
Even the quantity does not seem to be well defined.

b) Similar issue occurs in Lemma 1 - Neighboring datasets S and S' are created by first sampling S from P and then S' is obtained by replacing an arbitrary point in S by a random point from P*. Then sensitivity is defines as a max over pair of neighboring datasets - again S and S' are random samples, so what is the max over ?? If it is the worst case - why is the sampling coming in there ? Since it uses Corollary 1 - the main result inherits the same fundamental issues the has been pointed out above.

c) In theorem 2, {cal F}_a is an algorithm. What does it mean to add noise ? - Does it mean you add noise to the model parameters ?? - This is confusing at best.

Minor Issues:
1. Authors claim that connection between causality and privacy has not been explored (Page 2). Pls refer to https://arxiv.org/pdf/1710.05899.pdf where differential privacy itself is related to interventional effects in a system. This connection is very different from the scope of the current paper. However, the statement by the authors is strictly not true.

2. Why is the ground truth function f:X->Y (Section 2.2.) relevant when clearly you have distribution P (X,Y) and P*(X,Y) ?? We might as well define Loss with L (h(x), y) where (x,y) is drawn according to P. What f is never defined anywhere. Authors seems to mean the suggestion I just said in the paper. Authors could clarify. This confuses stuff in the proofs too.

3. Markov Blanket is not causal by any means in my opinion. It is just a minimal set of features conditioned on which Y does not depend on anything else. This only requires conditional independence tests to determine - a purely observational notion - in fact the markov blanket only depend on the moralized graph which does not change across the members of the equivalence class. So calling it causal is a bit confusing. If the features referred to least causal Parents - then still it would be consistent with the invariance in the Invariant Causal Prediction Literature (Peters et al 2016.) and would be causal.









**Experience Assessment:**

I have published in this field for several years.

**Review Assessment: Checking Correctness Of Derivations And Theory:**

I carefully checked the derivations and theory.

**Review Assessment: Checking Correctness Of Experiments:**

I assessed the sensibility of the experiments.

**Review Assessment: Thoroughness In Paper Reading:**

I read the paper thoroughly.

---

> ### Author Response · Authors · 2019-11-11
> **Response to Review 2**
>
> >1. Why is $h_{c,P}^{OPT} = h_{c,P*}^{OPT}$ ? This is claimed for any loss function L (not just Cross Entropy Loss) and for a generic hypothesis class $H_C$ (that depends only on Markov Blanket).
>
> We acknowledge that the proof will not work if the hypothesis class does not include the true $P(Y|X_C)$ (or equivalently, the true function $f$). Therefore, we have updated the proof to assume that the hypothesis class includes f. Even under this condition, there can be multiple associational models depending on the specific distribution P and thus it is possible that $h_{a,P}^{OPT} != h_{a,P*}^{OPT} $.
> As an example, consider a colored MNIST data distribution P where the classification task
> is to detect whether a digit is greater than 5 , and where all digits above 5 are colored with the same color. Then, under a suitably expressive class of models, the loss-minimizing associational model may use only the color feature to obtain zero error, while the loss-minimizing causal model will still use the shape (causal) features to obtain zero error. On any new P∗ that does not follow the same correlation of digits with color, we expect that the loss-minimizing associational model will now be different, probably that uses the shape features.
> Specifically, while we agree that the causal model will be one of the loss-minimizing associational models in P, it will not be the only one, and in general, associational models will not be able to distinguish between them if both minimize the loss equally. Thus, the best associational model over P can be different from the best associational model over P*. We have correspondingly updated the proof for Theorem 1 in the paper.
> On the choice of loss functions, we do have some restrictions on the loss function---we assume a symmetric loss function that follows the triangle inequality. We appreciate your point on the issue of general loss functions---can you provide some justification on why they may not work?
>
> >2.  a) What is the max operator over in equation 5?  Similar issue occurs in Lemma 1.
>
> We wanted to convey the maximum over all x and x' such that x is part of the training dataset ($x \in S$) and $x'$  is outside the training set but still follows the same causal labelling function (i.e., $y'=f(x_c')$). We have clarified this in the revised statement for Corollary 1. We have updated the statement of Lemma 1 to remove the notion of sampling. The proof remains the same.
>
> > c) In theorem 2, ${\cal F}_a$ is an algorithm. Does it mean you add noise to the model parameters ?
>
> Yes, the noise is added to the model parameters obtained as an output of the learning algorithm. Sorry for the confusion. We have clarified it in the paper.
>
> Minor Issues:
> > 1.  https://arxiv.org/pdf/1710.05899.pdf  shows explores causality and privacy.
>
> Thanks for the reference. We have edited our statement to emphasize the scope of our paper in light of the provided reference. We now say, "the connection of effect of causal learning to privacy is yet unexplored."
>
> > 2. Why is the ground truth function f:X->Y (Section 2.2.) relevant when you have distribution P (X,Y) and P*(X,Y) ?
> We have now defined $f$ in Theorem 1 statement. We agree that we can define the loss wrt y as $L(h(x), y)$, but adding $f$ provides some conceptual ease during the proof. We can argue that $f$ remains invariant across the two distributions, and that the causal model learns the $f$ successfully.
>
> > 3. Markov Blanket is not causal. If the features referred to least causal Parents - then still it would be consistent with the invariance in the Invariant Causal Prediction Literature (Peters et al 2016.) and would be causal.
>
> We consider a "causal" Markov Blanket that is derived from the causal graph, not the "associational" Markov Blanket that is typically derived from a particular data distribution. To clarify, the structural causal graph remains the same across two distributions P and P*, and our definition of the Markov Blanket is based on this causal graph.
> Each of the two distributions, P and P*, will have their own conditional probabilities, and thus different associational Bayesian networks and corresponding Markov Blankets. For example, in the colored MNIST example above, the color of the digit will also be included in the associational Markov Blanket in P, but is not a part of the causal Markov Blanket. Thus, to the extent that Markov Blanket is derived from a causal graph, we consider it causal. More generally, our goal is prediction, not causal inference. That is, we are interested in constructing a model using stable relationships between X and Y that generalize well. In some cases, that stable relationship may be between Y and its child (e.g., a disease and its symptom). In that case, we believe it is okay to construct a "causal" predictive model using the child of Y (e.g., using the symptom to predict the disease), as long as we are not including correlational features like Age or Income.

---

> > ### Comment · AnonReviewer2 · 2019-11-11
> > **Partial feedback to rebuttal points**
> >
> > I appreciate the revision and the feedback to address by concerns. I have issues with point number 1 still. I will come back to it shortly. Let me ask a quicker question to your answer on point 2.
> >
> > 2 - "We have now defined $f$  in Theorem 1 statement. We agree that we can define the loss wrt y as $L(h(x),y)$ , but adding  provides some conceptual ease during the proof. We can argue that  remains invariant across the two distributions, and that the causal model learns the  successfully. "
> >
> > I am looking at Theorem 1. " It says Let $f:X_C \rightarrow Y $ be the resultant invariant labeling function such that $y=f \left( X_C \right)$ " Can you mathematically say what it is in terms of $P (Y|X_C)$ ? Thats what I was looking for.  The reason I am asking is the following - usually in supervised learning, $Y$ is labeled according to $P(Y|X)$ meaning that the label is actually randomly drawn from the conditional. Now, the map estimate is usually (depends on the loss function $L$) $\argmax_y P(y|x)$ - this is what we use to label a test dataset. That does not imply that another dataset would be generated using a deterministic labeling function.

---

> > > ### Comment · AnonReviewer2 · 2019-11-11
> > > **Regarding MNIST color example - Invariance wrt Markov Blanket is violated**
> > >
> > > I agree that in practice a classifier trained would look for the easiest spurious correlation. I dont deny that. However, your paper is about optimal associational models without regard to any real world sub-optimal algorithm for obtaining a model. In your example,
> > >
> > > Shape -> digit -> color . Then for the digit (which is the label), according to Markov Blanket definition
> > > parent, children and parents of children are included. This means Markov Blanket includes (shape and color).
> > >
> > > So Shape and color both are in the markov blanket. Due to no association between color and digit in the test dataset, P(Y |Markov Blanket) is not the same - Infact Markov Blanket itself is different between P and P*. Your example does not satisfy assumptions of Theorem 1 ! Am I missing something ?

---

> > > > ### Author Response · Authors · 2019-11-12
> > > > **Response for MNIST color example**
> > > >
> > > > Thank you for your comment. We provide our explanation below.
> > > > On MNIST example: We are distinguishing between "causal" Markov Blanket that is derived from a structural causal graph, and the associational Markov Blanket that is derived from a probability distribution (or Bayesian network).
> > > >
> > > > In this example, here are the three graphs and corresponding Markov Blankets:
> > > > 1. Causal Graph: Shape -> digit (shape causes digit. MB = {Shape})
> > > >
> > > > 2. Associational Network, Train: Shape-> digit <--> Color (digit is probabilistically associated with shape and color, Associational MB ={Shape, Color}
> > > >
> > > > 3. Associational Network, Test: Shape-> digit ( no association of digit with color, Associational MB = {Shape})
> > > >
> > > > Thus, the causal graph and the causal MB remain the same across train and test distributions, satisfying the assumption in Thm 1. However, train and test do exhibit different probabilistic associations---we observe a correlation between Digit and Color in train dataset. This need not be causal---it could simply be due to selection effects while collecting the train dataset (e.g. for class 1, other colors are not sampled from a full master dataset with all colors).
> > > >
> > > > Under this setup, assuming that a model built using Color feature was equally predictive during training, we posit that even an optimal associational model may choose a function based on Color. Under the causal graph, we would still claim that shape causes the digit, and color happens to be correlated with the digit in a specific dataset.

---

> > > > > ### Comment · AnonReviewer2 · 2019-11-12
> > > > > **I disagree with the characterization**
> > > > >
> > > > > You said , for the train, $<=5$ (or $\geq 5$) is given the same color (or may be even noisily given a color with heavy bias).
> > > > >
> > > > > Then Color is generated looking at the digit and adding noise.
> > > > >
> > > > > Why is the causal graph limited to Shape->digit.
> > > > >
> > > > > Your generating mechanism is Shape -> digit-> color.
> > > > >
> > > > > So the Markov blanket has the color ??

---

> > > > > > ### Author Response · Authors · 2019-11-13
> > > > > > **Response: Generating process for a distribution may be different from the causal process**
> > > > > >
> > > > > > The short answer is that the generative mechanism for a particular data distribution is not always the causal mechanism. Based on our answer on the need for the definition of a causal Markov Blanket, we really need to find the causal mechanism to construct a causal MB. In causal inference literature, this phenomenon is known as selection bias when the data-generating mechanism has components that do not correspond to the causal mechanism. Since this is a fundamental question, let us respond by first clarifying the formal definitions, and then continuing with the colored MNIST example.
> > > > > >
> > > > > > Causal Mechanism: A causal mechanism can be formalized through a causal graph, where each edge $A \rightarrow B$ has a specific interventional interpretation: changing A will change B. Formally, causal mechanism specifies Pr(B|do(A)).
> > > > > >
> > > > > > In general, the same causal mechanism can be present in multiple data distributions P(A, B) and thus $P(B|A) \neq Pr(B|do(A))$ for every  distribution P.
> > > > > >
> > > > > >
> > > > > > In our continuing MNIST example, let us assume A=digit and B=color. If you only observe data from the $P$ distribution, then you can correctly detect $P(Color|Digit)$ as the data-generating process. You may also be tempted to declare that Digit causes Color, however, there is incomplete evidence to determine the causal mechanism as defined above. To determine a causal relationship, you have to ask the question whether changing Digit(A) will change the color (B)?
> > > > > >
> > > > > > In general, given only data from $P$, it is impossible to determine whether this $A \rightarrow B$ relationship is causal (without making other assumptions outside of the data). Outside of doing an actual intervention or experiment, our paper builds on recent work suggesting that we can use the invariance property of causal relationships---a causal relationship should be invariant across many different data distributions. In our continuing example, we found a $P^*$ where Color is no longer associated with Digit, and thus observing data from both $P$ and $P^*$  confirms that Digit->Color is not invariant, and thus not a causal relationship. Therefore, it should not be in included in the causal Markov Blanket (but is included in the associational Markov Blanket for P, see our response to associational versus causal Markov Blanket above).
> > > > > >
> > > > > > More generally, it is always possible to have a generating mechanism for a data distribution that is not a causal mechanism, and thus does not correspond to the causal Markov Blanket. Thus, in our paper (and theorem 1), we assume the existence of pre-specified Causal Graph based on outside domain knowledge.

---

> > > ### Author Response · Authors · 2019-11-12
> > > **Response to concerns of Point 2 on f**
> > >
> > > On definition of $f$: You raise a great point about $f$'s connection to P(Y|X)--thanks for this. For $f$, we are primarily following the structural causal model literature (Pearl 2009) that defines the value of a node variable in a causal graph in the form of a function of its parents. That is, $y=f(y_{parents} + \epsilon$ where $\epsilon$ is independent noise. Here, $f(y_{parents})$ can be considered as the expected value of P*(Y|Y_{parents}). Thus,
> > >
> > > $$ f(y_{parents}) = E[Y|y_{parents}] $$
> > >
> > > In this paper, we simplify the structural causal model to remove any independent noise term, thus following the domain adaptation literature (e.g., Mansour et al.) that assumes a deterministic $f$. Hence, we say $y =f(x_c)$ (and so, $f(x_c)=E[Y|x_c]$ with zero variance). Specifically, we also consider $x_c$ as the full Markov Blanket in addition to parents, as our goal is prediction only. We believe this setup will be relevant for classification problems where it is reasonable to assume a deterministic outcome given the inputs. That said, it should be possible to generalize to the case where $y=f(x_c) + \epsilon$ in future work.
> > >
> > > Mansour et al.:Domain Adaptation: Learning Bounds and Algorithms. Mansour, Mohri, Rostamizadeh (2009)

---

> > > > ### Comment · AnonReviewer2 · 2019-11-12
> > > > **This specification of f need to be in the Theorem 1 - still it it gives rise to more problems. Learnt causal models might be different between $P$ and $P*$**
> > > >
> > > > I would suggest the authors then to specify the definition of f in Theorem 1.  It seems to be a crucial detail.
> > > >
> > > > Suppose Y was Gaussian with mean $\mu(X_c)$ where $\mu$ is a mean function which is linear $\mu(x_c) = \theta^T x_c$ - while the variance is some $\Sigma(X_c)$. Then clearly your $f(x_c)= \theta^T x_c$ (by your definition)
> > > >
> > > >  Suppose, P(Y|X_C) is sampled from a Gaussian with the same mean $\mu(\cdot)$ but different variance functions $\Sigma_1(x_c)$ and $\Sigma_2(x_c)$ for two different parts of the domain $D_1$ and $D_2$ and you have one mixture of the domains for P and another mixture for P*.
> > > > By the way $P (Y|X_C)$ is invariant here over the entire domain.
> > > >
> > > > Suppose the loss function was something *other than* squared loss (say just $\ell_1$ for example)  and u train with this loss on the labels in the dataset, then even if you include all linear functions (that indeed captures your labeling function $f$), then again the invariance will not hold.
> > > >
> > > > So still $h_{c,P}^{OPT} \neq h_{c,P*}^{OPT}$ !
> > > >
> > > > I think this formulation may have fundamental problems.
> > > >
> > > > Now if you define $L$ with respect to $h$ and $f$ it seems like the label $y$ in the dataset is never used (since both $h$ and $f$ are dependent only on the point). So in what sense is it a good evaluation ?
> > > >
> > > > Let us assume as the authors claim that $f \in {\cal H}_C \subset {\cal H}$. $L(f,f)=0$ I assume. So wont the trivial solution in every case be $h^{OPT}=f$ ??
> > > > Since, $h^{OPT}$ is defined as $\argmin_{h} L(h,f)$ (as the authors define in Appendix I)

---

> > > > > ### Author Response · Authors · 2019-11-13
> > > > > **Response:  Our specification assumes a deterministic f that ensures same causal models between P and P***
> > > > >
> > > > > As we mentioned in our reply above, we assume a deterministic f. Thus the variance $\sigma_1(x_c)$ and $\sigma_2(x_c)$ will be zero. Then in your example of a Gaussian $P(Y|X_C)$, we would obtain $h_{c,P}^{OPT}=h_{c,P^*}^{OPT}$. Now we agree that even for an associational model, one of the optimal solutions will be $h_{a,P}^{OPT}=f$, but depending on the data distribution, there can be other equally optimal solutions that are not $f$, and the associational learning algorithm will have no way to distinguish between them (that is, it may pick the incorrect one even with infinite data). We make this exact point in our Theorem 1 proof (page 12, paragraph: "Associational Models").
> > > > >
> > > > > That said, we do acknowledge that Theorem 1 does not include the more general case where the true labelling function $f$ is not deterministic.
> > > > > When $f$ is non-deterministic, the result depends on a number of factors, including the loss function, and the relative scale of changes in P(X) (covariate shift) and the changes in P(Y|X) (concept drift). For instance, your example provides a good illustration of the issues with the choice of loss-function (squared loss works, but not l1). Similarly, the relative amounts of covariate shift and concept drift matter. For example, if $P^*(X) \approx P(X)$ but $P^*(Y|X)$ and $P(Y|X)$ vary by a lot, then a causal model will have lower error. But if $P^*(Y|X) \approx P(Y|X)$, $P^*(X)$ and P(X) vary by a lot, then it is not clear. (note that all of the above statements are for $X$, not $X_C$).
> > > > >
> > > > > Overall, however, we thought that $f$ being deterministic is a reasonable assumption to make, following the domain adaptation literature which also assumes a deterministic $f$.  Further, if one assumes that the variance in the true function $f$ is small enough or negligible, then all of the claims of Thm 1 should follow in practice. Thus, we chose the setting of a deterministic $f$ to simplify the proof and present the main conceptual argument. More generally, we believe that rather than a fundamental problem with our formulation, Theorem 1 provides fruitful ground for future work on generalizing to non-deterministic $f$, and the associated trade-offs between covariate shift and concept drift, and on sensitivity to choice of a loss function.
> > > > >
> > > > > Thank you for engaging in this discussion. For completeness, we will include the specific definition of f in Theorem 1 and highlight the issues when $f$ is non-deterministic in our paper.

---

> > ### Comment · AnonReviewer2 · 2019-11-12
> > **About Markov Blanket - what is the difference between associational Markov Blanket versus Causal Markov Blanket ?**
> >
> > "We consider a "causal" Markov Blanket that is derived from the causal graph, not the "associational" Markov Blanket that is typically derived from a particular data distribution. "
> >
> > Can you please provide a reference for this. Whats the difference between associational Markov Blanket and Causal Markov Blanket?
> >
> > Markov Blanket is the smallest subset of variables observed conditioned on which the rest of the variables become independent - It is a "purely" observational notion.
> > In a Bayesian network (Causal or Otherwise) it is the parents, children and co-parents.
> >
> > Now if you moralize the Causal Bayesian Network into an undirected graph, all neighbors of Y in the moralized undirected model will be in the Markov Blanket - this again can be determined by just CI tests - a purely observational notion. Just because in the Causal Bayesian Network, parents + children and coparents form the markov Blanket, it does not mean it is a function of the causal graph - it is a purely observational quantity - See reference  https://pdfs.semanticscholar.org/53a7/28fcf178f418a4fc3297f8ab0f04e12c5df7.pdf.  Please see definition 1 - what is causal about that definition ? It is a purely observational notion
> >
> > Why do you need the causal  graph to define the Markov Blanket ? There is an algorithm called IAMB (a Markov Blanket discovery algorithm) which works ONLY on observational data to find the Markov Blanket - again see reference https://pdfs.semanticscholar.org/53a7/28fcf178f418a4fc3297f8ab0f04e12c5df7.pdf.
> >
> > Another reference discussing Markov Blanket  (please see the definition in the Background section) - https://www.aaai.org/Papers/FLAIRS/2003/Flairs03-073.pdf.

---

> > > ### Author Response · Authors · 2019-11-13
> > > **Response: Causal versus associational Markov Blankets**
> > >
> > > A Markov Blanket is an observational notion only under two strong assumptions: Causal Markov Condition and Faithfulness. Pellet et al. provide a good description of these conditions and define a "perfect map" (Definition 6). This is also mentioned in the reference you shared, Flairs03-07.pdf.
> > >
> > > The Causal Markov Condition states that if two variables are d-separated (independent) in the causal graph, then they should also be independent in the data distribution.
> > > The faithfulness property states that the observed data distribution contains no independences between variables that do not follow from d-separation on the underlying causal graph.
> > >
> > > Together, they imply that a causal graph is a "perfect map" of a data distribution. (In)dependence in the graph corresponds to (in)dependence in the data distribution. In practice, however, these assumptions can be violated, and often are. Our paper addresses the setting when these assumptions can be violated.
> > >
> > > When the perfect map property is violated, then the Markov Blanket cannot be uniquely determined from a data distribution. Therefore, we introduced two kinds of Markov Blankets:
> > > 1. "Associational" MB, that are determined from a data distribution assuming a perfect map, and
> > > 2. "Causal" MB, that cannot be determined from the data distribution alone and is derived from domain knowledge.
> > >
> > > For instance, consider our continuing example of colored MNIST. Using a dataset from P, we will claim that {Shape, Color} is the MB for Digit. Using a dataset from $P^*$ that does not have the association between Color and Digit, we will claim that {Shape} is the MB.
> > >
> > > Now both cannot be correct. If the true causal graph is Shape->Digit->Color, then $P^*$ violates faithfulness assumption and thus the Markov Blanket estimated from $P^*$ is incorrect. If the true causal graph is instead Shape->Digit, then $P$ violates the Causal Markov Assumption and thus the Markov Blanket from $P$ is incorrect.
> > >
> > > To reduce the resulting confusion in nomenclature, we say that the MB derived from observed data distribution is an "associational" MB and can vary across distributions. And the MB derived from a causal graph is the "causal" MB and stays invariant across distributions.
> > >
> > > Thanks again for these questions. To summarize, we need a causal notion when a single data distribution cannot uniquely identify the Markov Blanket. These will help us justify our nomenclature in the revised paper.
> > >
> > > Pellet et al. Using Markov Blankets for Causal Structure Learning. JMLR 2008.

---

### Author Response · Authors · 2019-11-11
**General Comment & Updates to paper**

We thank the reviewers for their feedback. We provide individual response to each of the reviewers. We outline our main contributions again as follows.

	1. Our goal in the paper is not to learn a causal structure, but rather evaluate the predictive accuracy of models as the feature distribution changes specifically, we mean domain shift in $P(X)$.
	2. We want to demonstrate the privacy guarantees that causal learning provide by empirically demonstrating their robustness to membership inference attacks.
	3. Our aim is to make the community aware of the privacy benefits of causal learning and consider the importance of using causal features for predictions in sensitive applications.

We summarize the changes made in the updated version of the paper:
	1. We updated statement and proof of Theorem 1 to clarify hypothesis class
	2. We updated statements for Corollary 1 and Lemma 1 to clarify max operator
	3. We clarified our claim of connecting causal learning  and privacy for membership attacks
	4. We have clarified invariance of causal model in abstract
        5. We have clarified the addition of noise to the trained model parameters for making them differentially-private.

---

### Decision · Program_Chairs · 2019-12-19

**Decision:**

Reject

**Comment:**

Maintaining the privacy of membership information contained within the data used to train machine learning models is paramount across many application domains.  Moreover, this risk can be more acute when the model is used to make predictions using out-of-sample data.  This paper applies a causal learning framework to mitigate this problem, motivated by the fact that causal models can be invariant to the training distribution and therefore potentially more resistant to certain privacy attacks.  Both theoretical and empirical results are provided in support of this application of causal modeling.

Overall, during the rebuttal period there was no strong support for this paper, and one reviewer in particular mentioned lingering unresolved yet non-trivial concerns.  For example, to avoid counter-examples raised the reviewer, a deterministic labeling function must be introduced, which trivializes the distribution p(Y|X) and leads to a problematic training and testing scenario from a practical standpoint.  Similarly the theoretical treatment involving Markov blankets was deemed confusing and/or misleading even after careful inspection of all author response details.  At the very least, this suggests that another round of review is required to clarify these issues before publication, and hence the decision to reject at this time.